## Perspective

neuroscience/cognition/behaviour

insula, interoception, active inference, feelings, depression

**Author for correspondence:**
Alan S. R. Fermin
e-mail: fermin@hiroshima-u.ac.jp

# An insula hierarchical network architecture for active interoceptive inference

Alan S. R. Fermin[1], Karl Friston[2] and Shigeto Yamawaki[1]

[1]Center for Brain, Mind and Kansei Sciences Research, Hiroshima University, Hiroshima, Japan
[2]The Wellcome Centre for Human Neuroimaging, UCL Queen Square Institute of Neurology, London, England

ASRF, 0000-0003-4014-6365; KF, 0000-0001-7984-8909;
SY, 0000-0003-4263-6771

In the brain, the insular cortex receives a vast amount of interoceptive information, ascending through deep brain structures, from multiple visceral organs. The unique hierarchical and modular architecture of the insula suggests specialization for processing interoceptive afferents. Yet, the biological significance of the insula's neuroanatomical architecture, in relation to deep brain structures, remains obscure. In this opinion piece, we propose the Insula Hierarchical Modular Adaptive Interoception Control (IMAC) model to suggest that insula modules (granular, dysgranular and agranular), forming parallel networks with the prefrontal cortex and striatum, are specialized to form higher order interoceptive representations. These interoceptive representations are recruited in a context-dependent manner to support habitual, model-based and exploratory control of visceral organs and physiological processes. We discuss how insula interoceptive representations may give rise to conscious feelings that best explain lower order deep brain interoceptive representations, and how the insula may serve to defend the body and mind against pathological depression.

## 1. Introduction

The human brain comprises various anatomical regions, specialized for diverse functions, such as language, problem-solving, decision-making, memory, motivation, inhibitory control, emotion, motor control and social cognition [1]. Recent studies have sought to understand how such highly complex psychological processes, and pathological states, e.g. depression, are influenced by interoception, the sensation of information ascending to the brain from visceral systems, physiological processes and circulating chemicals under the control of the autonomic nervous system (ANS) [2–11]. The ANS is responsible for processing and transmitting interoceptive information to the

brain from the visceral organs that maintain survival functions, including the gastrointestinal, cardiovascular, respiratory, thermoregulatory, hormonal and immune systems [3,4]. However, the functions for which cortical brain regions receiving interoceptive afferents are specialized, and when such functions are recruited for survival, remain obscure.

The brainstem is the region that receives direct interoceptive information ascending from the visceral systems [1]. The brainstem contains several nuclei that receive interoceptive information, such as the nucleus tract solitary (NTS), the medullary reticular formation, the parabrachial nucleus (PBN) and the periaqueductal gray area (PGA). These nuclei are thought to generate innate, hard-wired, visceral and hormonal responses to maintain all physiological survival functions and to cope with demands imposed by the body and the environment [1,12]. In the cerebral cortex, the insula and anterior cingulate cortex (ACC) are the main targets of interoceptive afferents arriving from the visceral systems through thalamic nuclei [3,5,13–17]. However, while the ACC is a predominantly agranular structure, i.e. it lacks a granular layer IV, the insula has a more complex organization with a topographic neuroanatomical representation of visceral processes and three sub-regions with distinct levels of laminar granularity, distribution of acetylcholinergic receptors, and patterns of local, cortical, subcortical and brainstem connectivity [13,17–19] (figure 1). These cytoarchitectonic and anatomical connectivity features suggest a central position and specialization of the insular cortex in processing interoceptive information. This convergence of cortical and interoceptive information upon the insular cortex has led to influential hypotheses concerning its potential roles in interoceptive prediction [14], information integration for awareness [20,21], emotional awareness [22], interoceptive inference and emotion [23,24] and error-based learning of feelings [25]. Despite the elegance and appeal of these hypotheses, no one has yet explained why the brain needs a cortical insular representation of visceral processes, given that the various brainstem nuclei and other subcortical systems, e.g. hypothalamus, are sufficient to generate all necessary visceral and physiological adjustments to maintain the body's survival functions. These hypotheses have also suggested that insular functions are supported by its interactions with other brain regions [14,21,22,25]. However, no details have been suggested regarding mechanisms by which such interactions occur, nor their specific roles in interoception. Thus, it remains unclear how insular functions interact with and are modulated by input from cortical, subcortical and neuromodulatory systems, and whether such interactions underwrite survival.

In this opinion piece, we turn to studies of insula cytoarchitectonic organization, neuroanatomical connectivity and recent theoretical formulations of brain function, such as allostasis, predictive coding, and active inference, to put forward the Insula Hierarchical Modular Adaptive Interoception Control (IMAC) model. The IMAC model proposes that the hierarchical and modular organization of the insular cortex, supported by its reciprocal connections with the prefrontal cortex (PFC) and the striatum, and modulated by the dopaminergic and acetylcholinergic systems, mediates (i) context- and behaviour-dependent control and learning of visceral and physiological responses and (ii) higher-order representation of conscious interoceptive feelings, which are built upon basic emotions and their underlying visceral processes.

According to the theory of active inference, the brain uses internal generative models, acquired through experience or by mental simulation, to continuously generate descending or top-down predictions of expected sensory data [26–29]. In active inference, the goal of the agent is to find optimal action policies, e.g. rules or strategies for quick selection of actions, muscle activation patterns, decisions and social behaviors in a given context, that minimize free-energy, or prediction errors, between predicted and actual sensory input generated by the agent's interactions with, or sampling of the environment, e.g. quality of social interactions at home or in public, street navigation while driving or walking, selection of healthy food, learning to play a musical instrument, whether to dribble or pass the ball while playing basketball, an infant learning to walk on a slippery or rough surface [30,31]. The theory of allostasis proposes a similar predictive process for regulation of visceral organs and physiological states of the body [7,32–34]. The allostasis model suggests that the brain and visceral systems use innate or learned prior knowledge of physiological states, e.g. glucose levels and heart rate, to predict future visceral and physiological states, thereby pre-emptively precluding deviations or prediction errors from homeostatic setpoints. Recent studies have unified the concepts of active inference and allostasis under the umbrella of active interoceptive inference to suggest that the brain also creates and stores generative interoceptive models of the internal milieu of the body and uses such interoceptive models to explain ascending interoceptive signals and to generate descending interoceptive predictions to regulate and achieve desired states of the visceral organs and physiological processes, such as heart rate, hormone release, activation of the immune system and energetic metabolism [10,14,23,29,35,36]. According to the active interoceptive inference approach, the

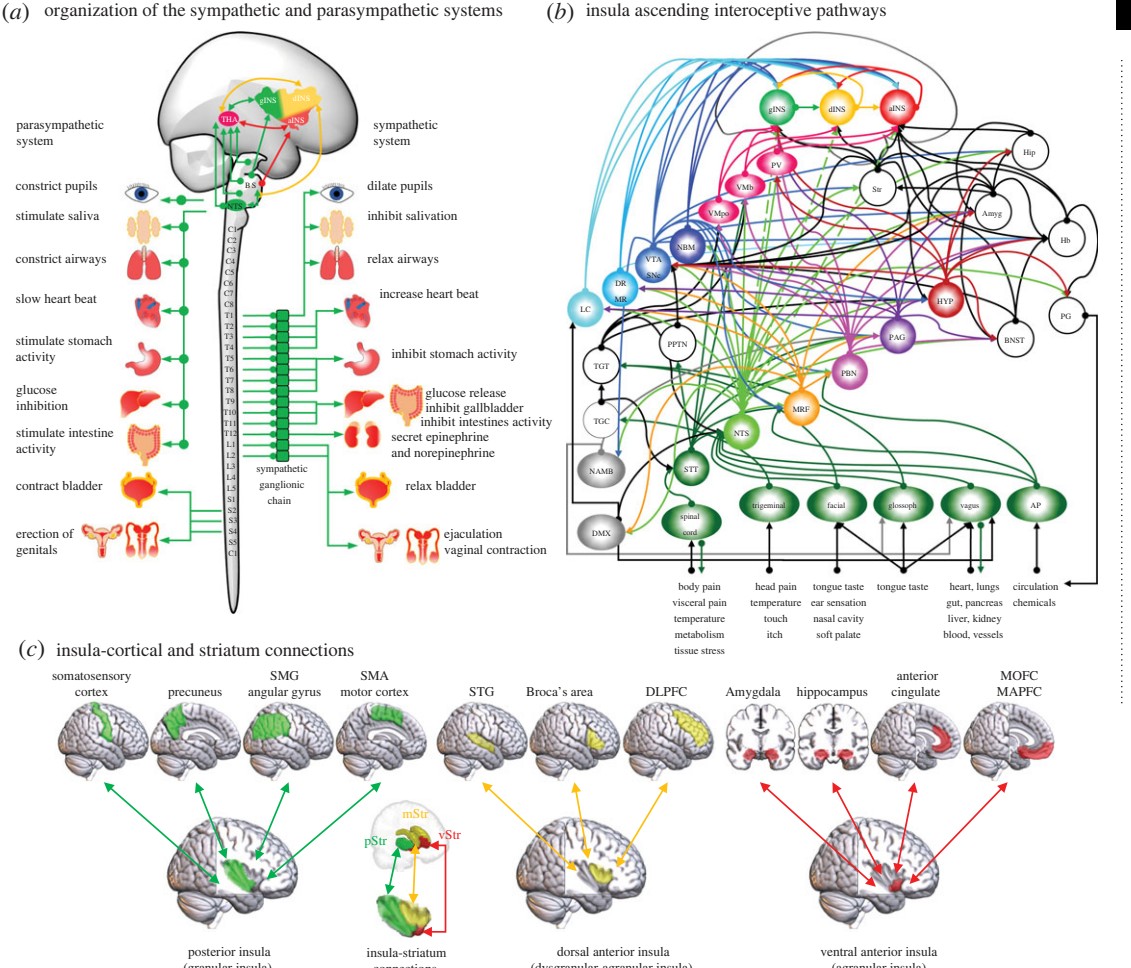

**Figure 1.** (a) Basic organization of neural pathways linking the parasympathetic and sympathetic branches of the autonomic nervous systems to the insular cortex and their effects on visceral functions. (b) Schematic diagram of ascending interoceptive pathways to the insula. (c) Distinct cortical and striatum connections that predominantly target each insula modular cytoarchitecture. gINS (granular insula), dINS (dysgranular insula), aINS (agranular insula), VMb (ventromedial basal thalamus nuclei), VMpo (ventromedial posterior thalamus nuclei), PV (paraventricular thalamus nuclei), HYP (hypothalamus), PBN (parabrachial nucleus), PAG (periaqueductal gray), NTS (nucleus tractus solitarius), AP (area postrema), NAMB (nucleus ambiguus), MRF (medulla and reticular formation), STT (spinothalamic tract), TGC (trigeminal complex), TGT (trigeminal thalamic tract), PPTN (pedunculopontine tegmental nucleus), BNST (bed nucleus of the stria terminallis), LC (locus coeruleus), DR-MR (dorsal raphe and median raphe), VTA-SNc (ventral tegmental area and substantia nigra pars compacta), NBM (nucleus basalis of Meynert), Str (striatum), Hip (hippocampus), Amyg (amygdala), Hb (habenula), pStr (posterior striatum), dStr (dorsomedial striatum), vStr (ventral striatum), SMG (supramarginal gyrus), SMA (supplementary motor area), STG (superior temporal gyrus), DLPFC (dorsolateral prefrontal cortex), MOFC (medial orbitofrontal cortex), MAPFC (medial anterior prefrontal cortex).

goal of the agent and the brain is to find optimal interoceptive policies, e.g. visceral and physiological response patterns that can be quickly selected for implementation in a given context, that minimize interoceptive prediction errors between predicted and actual interoceptive input arriving from visceral and physiological systems. Interoceptive policies are acquired by sampling visceral responses occurring at a given time and context, e.g. heart rate and hyperventilation, e.g. lung inflation while running, breathing speed, stomach motility and pain after a meal, bladder dilation with urine production, and other physiological processes, e.g. decrease in glucose level with increased hunger, hormones released after physical activity or psychological stress, immune molecules and inflammatory processes following tissue stress, body temperature changes in a cold or hot day.

Based on the allostatic reading of active inference [23,29,37,38], the IMAC model hypothesizes that the hierarchical modular cytoarchitecture of the insular cortex, supported by its parallel neural networks with the PFC and striatum-dopaminergic and acetylcholinergic systems, is specialized in higher-order interoceptive inference, herein called *metaception*. That is, it specializes in construction of cortical

representations of lower-order, innate, interoceptive representations, herein called *mesaception*, located in subcortical structures, i.e. amygdala and hypothalamus, and brainstem nuclei. In the IMAC framework, metaception is regarded as an evolved cortical capacity to generate flexible higher-order interoceptive predictions that are concurrently computed with action predictions that seek to maximize an individual's long-term fitness in interactions with the environment. In other words, metaception is the perceptual synthesis (cf., Bayesian belief updating) that furnishes high-level representations (cf., feelings), which can predict lower level interoceptive representations and accompanying responses in the motor and autonomic domain.

The main premise of the IMAC model is that adaptive behaviour of animals and humans revolves around acquisition not only of action policies, such as stimulus-action mappings or social interaction strategies that maximize rewards, survival and reproduction [39,40] but also the acquisition of interoceptive policies that are needed to maintain the body's physiological and visceral survival functions and concurrently to support mental processes and implementation of action policies, eventually leading to mapping or binding of action-interoception policies. Take the wake-sleep cycle as a simple example of an action-interoception policy mapping in which the brain uses interoceptive policies to generate visceral responses appropriate for wake-sleep behaviors: humans display higher blood pressure, increased heart rate, fast metabolism, reduced melatonin production and increased cortisol release in the waking period when the level of physical activity is higher than in the sleep period when the physiological demands of the body are highly diminished [41]. To foreshadow the conclusions below, IMAC conceives of active interoceptive inference as analogous to motor control, where action is realized by motor (resp. autonomic) reflexes that resolve proprioceptive (resp. interoceptive) prediction errors. These reflexes depend upon descending predictions or setpoints that are elaborated in deep hierarchical structures over increasing temporal scales.

In order to lay a foundation for a more detailed explanation of the IMAC model, this opinion starts with a brief overview of neural pathways linking the ANS to the insular cortex, followed by a description of the modular, hierarchical and cytoarchitectonic organization of the insula, as well as the parallel neuroanatomical networks linking the insula with the PFC and striatum. We will also describe neuromodulatory roles of acetylcholine within insula modular structures, and of dopamine on insula-PFC-striatum networks. We then introduce the IMAC model and consider how it mechanistically explains active interoceptive inference, the emergence of higher-order interoceptive representations, and possibly of conscious feelings: cf. [42]. We conclude by identifying directions in which concepts proposed by the IMAC model may be used to understand interoceptive dysfunctions observed in mood disorders, with a special focus on depression.

# 2. Insula neural architecture, cortical and subcortical connections

The ANS is organized into two main systems, the sympathetic nervous system (SNS) and parasympathetic nervous system (PNS), which convey interoceptive information about the physiological state of the body and visceral organs to the insular cortex and back to the viscera [3,4] (figure 1a). Interoceptive information, such as glucose levels, blood oxygenation and osmolarity, as well as interoceptive afference from the viscera located in the thorax and abdomen, including heart contraction, lung inflation and deflation, blood vessels and kidneys, reaches the insula through the brainstem via the PNS. The SNS transmits interoceptive information related to visceral pain, somatic pain, skin pressure, e.g. touch, tissue stress, metabolism and body temperature [3,4,20]. Molecular interoceptive information, including nutrients, transport of blood gases (oxygen and carbon dioxide) and concentration and regulation of ions in neural tissue, which allow maintenance of neural homeostasis, e.g. synaptic plasticity, development and preservation of neural structure, also reaches the brainstem and insular cortex through the vascular system and the blood–brain barrier [3,43–45].

Cell bodies of SNS afferent neurons are located in the dorsal roots of the spinal cord, whereas cell bodies of PNS afferent neurons bypass the spinal cord and are located in cranial sensory ganglia [3,4]. SNS interoceptive afferents ascend the spinal cord via the spinothalamic tract to synapse onto brainstem nuclei or directly onto thalamic nuclei before reaching the insular cortex. By contrast, PNS interoceptive afferent fibres ascend from visceral systems via cranial nerves e.g. glossopharyngeal and vagus, and synapse onto brainstem nuclei before reaching the thalamus and from there to the insula (figure 1b). The brainstem and insula receive interoceptive information from the head via the facial and trigeminal cranial nerves. In general, the SNS, PNS, facial and trigeminal nerves synapse onto the brainstem nucleus tractus solitary (NTS), making it the most prominent deep brain hub for integration

and relay of interoceptive information throughout the brain and back to the visceral systems [3,5,12,46]. The NTS receives significant chemical interoceptive signals from the area postrema (AP), a circumventricular organ located in the brainstem, which lacks a blood–brain barrier and senses chemical substances in the cerebrospinal fluid and circulation, e.g. hormones, immune molecules, that modulate visceral functions and behaviour [12,47]. The main visceral brainstem targets of the NTS are the AP, medullary reticular formation, PBN and PAG that form multiple, complex, parallel neuroanatomical pathways connecting with the hypothalamus and other subcortical, allocortical and forebrain areas, such as the amygdala, hippocampus, nucleus accumbens, habenula, bed nucleus of the stria terminalis and the pineal gland. Evidence from animal studies also suggests that the NTS sends direct projections to the insular cortex [46,48]. The majority of ascending interoceptive information originating from both PNS and SNS converge onto thalamic nuclei (ventromedial posterior nucleus, ventromedial basal nucleus, paraventricular nucleus) to reach, predominantly, the granular portion of the insular cortex [20,49]. Finally, NTS interoceptive signals can quickly exert influences over not only the insular cortex but the whole cerebral cortex via its direct projections to neuromodulatory systems that bypass other brainstem nuclei, including the dopaminergic system (ventral tegmental area and substantia nigra pars compacta), the noradrenergic system (locus coeruleus), the serotonergic system (dorsal raphe and median raphe) and cholinergic system (nucleus basalis of Meynert) (figure 1b) [46].

As described above, various types of visceral information reach the granular region of the insular cortex. The granular insula (gINS) is one of the three insular modular structures identified based on expression of cellular layer IV, containing predominantly granular cells: the agranular insula (aINS), located in a ventral anterior position and lacking a granular layer IV, the gINS, located in the most dorsal posterior position, with a fully developed layer IV, and the dysgranular insula (dINS), located between the aINS and gINS, but with an underdeveloped layer IV [19,50–55]. The modular organization of the human insular cortex (posterior and anterior segments) is already present in neonates [56]. Other studies have suggested a more detailed parcellation of the modular organization of the insula, sometimes sub-dividing the insular cortex into 31 sub-regions [13,49,54,55,57–60]. However, since the precise roles of these finer insular parcellations have yet to be determined, for simplicity, the IMAC model considered here will focus on the general three-insula modular cytoarchitecture (gINS, dINS and aINS) and candidate roles for these modules in processing interoceptive information.

Local anatomical connectivity among the three insula sub-regions is very distinct, with reciprocal connections between the gINS and dINS and between the dINS and aINS, and modest aINS fibre output to the gINS [50–52]. This pattern of connectivity suggests a hierarchical organization for interoceptive information processing within the insular cortex that is similar to the hierarchical organization observed in other sub-cortical and cortical systems that subserve perceptual, action control and higher-order cognitive processes, including the basal ganglia, and the visual, temporal and prefrontal cortices [61–71].

The modular organization of the insula is also supported by its parallel anatomical connectivity with the thalamus, brainstem nuclei (figure 1b), cortical regions and striatum (figure 1c), and by electrophysiological stimulation and functional neuroimaging studies in monkeys and humans. In monkeys, the aINS, dINS and gINS make bidirectional connections with thalamic nuclei, from which they receive interoceptive information [72]. Cortical regions involved in primary sensory (somatosensory and auditory cortices), motor (primary and supplementary motor areas) and environmental information processing (superior and inferior parietal cortices) project predominantly to the gINS [13,73]. It is important to note here that the inferior parietal cortex contains the supramarginal and angular gyri that form the temporo-parietal junction (TPJ), a cortical complex implicated in social cognition [74–79]. The dINS, on the other hand, receives anatomical input from the dorsolateral prefrontal cortex (Brodmann areas 45 and 46) [13,50–52,80]. By contrast, the aINS has predominant connections with the ACC, ventral anterior PFC, ventromedial orbitofrontal cortex, amygdala and hippocampal complex [13,50–52]. The insula also forms distinct parallel connections with the striatum: the aINS makes connections primarily with the ventral striatum (vStriatum), the dINS with the dorsomedial striatum (mStriatum) and the gINS with the dorsolateral posterior striatum (pStriatum) [19]. Striatum sub-regions also receive topographical input from the dopaminergic system, with the vStriatum receiving its main dopaminergic input from the ventral tegmental area (VTA), and the mStriatum and pStriatum from the medial and ventro-lateral substantia nigra complex, respectively [81]. Neuroanatomical evidence also suggests that each insula module has direct projections to brainstem visceral motor nuclei that modulate visceral and physiological processes, such as gastric functions, heart rate, blood pressure, pain and hormone secretion [82–91].

The insula contains topographic interoceptive or viscero-sensory maps located predominantly in its granular sub-region, extending in a posterior to anterior direction, that represent vestibular, nociceptive, thermoreceptive, visceral and gustatory information [13,49]. Stimulation studies in humans have revealed a similar topographic representation of pain, thermal, somatosensory, visceral, vestibular and gustatory information [92–95], although somatosensory information seems to be represented throughout the insular cortex [96,97]. Human neuroimaging studies also show a functional modular organization that maps onto the anatomical modular organization of the insula, with viscero-motor information represented in the gINS, emotional and motivational information represented in the ventral aINS, and cognitive information represented in the dorsal anterior insula, including the dINS [98–103]. Although here we use the multiple ascending interoceptive neuroanatomical pathways and their topographic representation onto the insular cortex as an important feature of the IMAC model, these pathways are not entirely anatomically segregated and non-overlapping in their insular representation. For example, human neuroimaging experiments have shown that insular representation of heart, stomach and bladder overlaps with representation of gustatory information [104–106]. The functional significance of overlapping neuroanatomical and functional insular interoceptive representations needs to be investigated in future studies.

Finally, the insular cortex has anatomical connections with the basal forebrain nuclei that contain cholinergic neurons, with stronger cholinergic efferents on the anterior aINS [107–110]. This pattern of insula-cholinergic connectivity is also linked with a progressive reduction of muscarinic acetylcholine receptors (rACh) along the insula ventral anterior to dorsal posterior axis, with a higher density of rACh in the aINS and a lower density in the gINS [50–52].

# 3. Insula hierarchical modular adaptive interoception control

In order for the insular cortex to promote influences over the visceral organs and physiological systems, the functions of the visceral organs and physiological systems should have plastic properties, i.e. to be malleable and flexible for changes. The idea of adaptable, flexible visceral functions has been extensively studied by psychophysiologists, who have used classical and operant conditioning paradigms to demonstrate that humans and animals can not only learn to generate anticipatory visceral responses that predict reward or punishment, but also to voluntarily generate visceral responses to achieve rewards or avoid punishments [111–118]. A well-known example of adaptive interoception control is Ivan Pavlov's classical conditioning experiments showing anticipatory salivation in dogs to reward-predicting cues and no salivation in response to non-reward-predicting cues [119]. Cardiovascular conditioning studies have reliably demonstrated that humans can voluntarily learn to increase or decrease heart rate and blood pressure [111,114,115,117]. Other studies with humans and animals have also observed anticipatory and voluntary control of visceral responses, including heart rate, blood pressure, blood volume, breathing, gastrointestinal function, bowel control, pupil dilation, electrodermal activity, body temperature, immunosuppression and blood oxygenation level [112,113,120–128]. Overall, these psychophysiological studies suggest that learning of visceral responses, at least for those which humans can exert voluntary control, may follow similar principles of adaption observed in motor behaviour, such as stages of learning and change in behavioural control, effect of prior knowledge, transfer of learning or generalization, efficiency of feedback, effector specificity and awareness of the learned visceral response [113,114,117,129–131].

An important question that emerges, following the successful demonstration by psychophysiologists that functions of visceral organs and other physiological processes can be modulated by experience is, 'How can the cerebral cortex, specifically the insular cortex, use interoceptive information to create generative interoceptive models that can explain the ascending interoceptive signals and regulate visceral functions in response to the many demands imposed by the environment and body?' Predictive coding has been proposed as a mechanism for learning of internal models by the cerebral cortex [66,132,133]. Ascending interoceptive inputs from the ANS and the pattern of anatomical connectivity among the insular modules suggest that interoceptive information first reaching the gINS from thalamic nuclei is then forwarded to the dINS, and from the dINS to the aINS. In contrast, backward, descending connections propagate information from the aINS through the dINS back to the gINS. These forward and backward connections endow the insular cortex with a neuroanatomical architecture suited to implementing predictive coding, where forward connections generate and convey ascending interoceptive prediction errors that inform the brain about inconsistencies in functioning of visceral systems and backward connections generate and convey descending

interoceptive predictions to regulate and correct identified inconsistencies in visceral functions [66,132]. In this hierarchical insular architecture, the gINS represents the lowest level of the hierarchy, the dINS is an intermediate level, and the aINS sits atop the hierarchy. The gINS is in a position to generate low-order interoceptive predictions and to compute interoceptive prediction errors by comparing those predictions with real-time interoceptive afferents from the ANS. The dINS generates intermediate-order interoceptive predictions and computes interoceptive prediction errors based on forward signals arriving from the gINS. Finally, the aINS generates higher-order interoceptive predictions and interoceptive prediction errors by computing the difference between its predictions and forward signals arriving from the dINS. Evidence that the insula is involved in computation of predictions or more specifically in generation of interoceptive predictions and uses Bayesian belief updating is still scarce, but has started to emerge from human and animal experiments [134–143].

The insula predictive coding mechanism elaborated above, although detailed, is not entirely new, as previous works have already proposed how active inference and predictive coding may be used for interoceptive inference [14,23,35] and how interoceptive information is processed in the posterior and anterior insula [14,20,21]. Anatomical studies demonstrating the existence of parallel networks linking the PFC and striatum [61,144,145] have paved the way to understand cognitive, emotional and motivational functions and dysfunctions of the basal ganglia, previously thought to be exclusively involved in motor control [64,103,129,146–155]. Oriented by these network approaches and in contrast to the earlier insula models, the IMAC model, however, suggests that insula active interoceptive inference functions can be better understood in light of its parallel connections with PFC sub-regions, namely the dorsolateral PFC (DLPFC), the ventromedial PFC (VMPFC), the supplementary motor area (SMA), with the striatum and neuromodulatory input from the dopaminergic and acetylcholinergic systems (figure 2). Earlier models also focused on the aINS-amygdala-brainstem network as the main pathway by which the insula affects visceral control [14,49,72]. However, the use of a single neural pathway by the central nervous system for visceral control may be insufficient and ineffective for maintenance and orchestration of all physiological needs and visceral organs of the body. Thus, the IMAC model proposes that the hierarchical and parallel insula-PFC-striatum networks offer several advantages for generation of interoceptive predictions in a context- and experience-dependent manner.

The PFC and striatum sub-regions form anatomical parallel loops specialized for processes implicated in adaptive behaviour, such as decision making, learning, emotion, motivation and sequential behaviours [129,144,146–149,156–159]. The VMPFC-vStriatum and DLPFC-mStriatum loops are recruited in early stages of learning, when behaviour is erratic, guided by external reward signals and require attention, but their activities diminish as learning progresses and behaviour becomes automatic, fast and less error-prone [129,148,149]. Distinct neuroanatomical components of the anterior PFC and VMPFC-vStriatum loop are also implicated in novel learning and decision-making by means of exploration and motivated behaviour [150,151,160,161], whereas the DLPFC-mStriatum loop implements a model-based decision-making strategy by using internal models of the environment or action representations to predict future outcomes of hypothetical actions [150,152,153,162]. By contrast, activity of the SMA-pStriatum loop increases after repeated experiences, when behaviours become automatic and habitual [129,154]. The SMA-pStriatum loop is implicated in sequential motor memory and habitual behaviour [129,148–150,163]. Decisions and action predictions generated in the VMPFC, DLPFC and SMA are sent to their striatal targets, where their signals are modulated and evaluated by dopaminergic confidence signals from distinct populations of dopamine neurons [30,61,164–166].

The IMAC model hypothesizes that the metaceptive functions of the insular modules follow similar functional specializations as those observed in PFC-striatum loops. Under this view, while the PFC-striatum-dopamine loops support learning and optimization of action selection and other cognitive, emotional or decision-making processes, the insula-striatum-dopamine loops are concurrently seeking to generate optimal interoceptive predictions to generate visceral responses necessary to achieve the physiological demands of desired actions, behaviours, motivations and mental processes. As an example of this action-interoception mapping, we have cited earlier the physiological changes observed in humans during wake-sleep cycles, such as changes in activity of neuromodulatory systems in the brain, e.g. higher serotonin and lower acetylcholine, increased heart rate, increased energetic metabolism, increased body temperature, increased respiration and decreased plasma melatonin during the waking period than during the sleep period [41,167,168]. During exercise, relative to the resting state, there are numerous physiological responses generated by visceral systems to support muscle performance, such as increased consumption of oxygen, increased cardiovascular, hormonal, metabolic, sweating and thermal regulatory responses [169–172]. There also are many daily situations in which the body generates physiological and visceral responses, such as increased heart

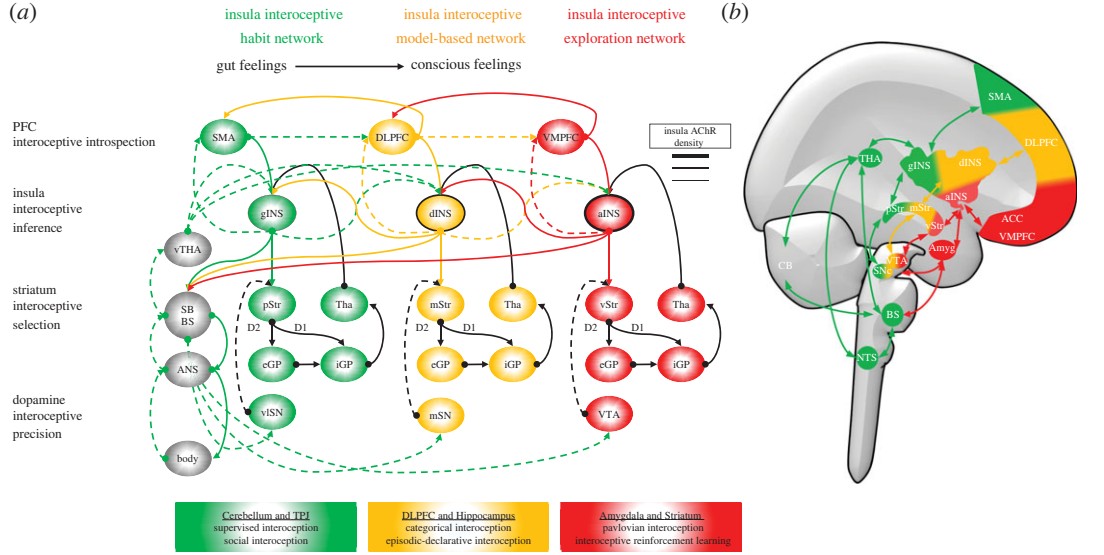

**Figure 2.** (*a*) Hierarchical insula, prefrontal cortex and striatum parallel networks for active interoceptive inference. In the IMAC model, parallel neuroanatomical networks linking the insula with the PFC and striatum, supported by the acetylcholinergic and dopaminergic systems, are specialized for distinct functions related to active interoceptive inference, such as interoceptive introspection, interoceptive prediction, interoceptive selection, interoceptive precision and interoception flexibility. The insula-PFC-striatum parallel networks are hierarchically organized as follows. The lower level comprises the gINS-SMA-pStr network specialized for storage and quick generation habit-like interoceptive predictions. The mid-level comprises the dINS-DLPFC-mStr network specialized for model-based interoceptive predictions, and the highest level comprises the aINS-VMPFC-vStr network specialized for interoceptive exploration. Other neural systems provide additional functions to the interoceptive predictions of these networks, such as supervised interoceptive predictions by the cerebellum, social interoception by the TPJ, episodic interoception and declarative interoception by the hippocampus, and Pavlovian interoception conditioning by the amygdala. The dashed line represents ascending interoceptive prediction errors. The continuous lines represent descending interoceptive predictions. The black lines represent connections within the striatum circuitry. (*b*) Schematic representation of the neuroanatomical location of the brain structures supporting insula active interoceptive inference. SMA (supplementary motor area), DLPFC (dorsolateral prefrontal cortex), VMPFC (ventromedial prefrontal cortex), ACC (anterior cingulate cortex), gINS (granular insula), dINS (dysgranular insula), aINS (agranular insula), pStr (posterior striatum), mStr (dorsomedial striatum), vStr (ventral striatum), eGP (external globus pallidus), iGP (internal globus pallidus), THA (thalamus), vlSN (ventrolateral substantia nigra complex), mSN (medial substantia nigra complex), VTA (ventral tegmental area), SNc (substantia nigra pars compacta), CB (cerebellum), NTS (nucleus tractus solitarius), D1 (dopamine receptor of the direct pathway), D2 (dopamine receptor of the indirect pathway), BS (brainstem), SB (subcortical brain regions, e.g. amygdala), vTHA (visceral thalamus: VMpo, VMb, PV [figure 1]).

rate, blood pressure, skin conductance, pupil dilation, in anticipation of aversive or reward predicting cues, public speech, social interactions and physical exercise [134,173–175]. Adaptive cardiovascular responses in humans are also observed in space flights, subaquatic diving, profession type and season of the year, in athletes of different sport modalities, and in response to the demands of cognitive and emotional tasks [176–180]. Without such visceral and physiological adjustments, it would be impossible to perform successful movements, have a good night of sleep, react appropriately to the demands of the environment or prepare the body and plan behaviors to anticipated stress.

Then what are insula-PFC-striatum parallel networks specialized for? The IMAC model suggests that the posterior gINS, which receives direct visceral input and forms a network with the SMA and pStriatum, is specialized to generate habitual interoceptive predictions using visceral-based representations, predictions realized by autonomic reflexes. According to this view, the posterior gINS stores interoceptive trajectories, which can be readily used as interoceptive policies, especially in well-learned environmental situations, for fast generation of interoceptive predictions that evoke visceral responses. A gINS interoception policy is selected by dopaminergic signals arriving from the ventro-lateral SN complex onto the pStriatum that evaluate the degree of confidence in gINS interoceptive predictions. This fits comfortably with the role of dopamine as encoding precision or salience of action-pointing representations [166].

There are environmental contexts and demands, however, for which gINS interoceptive policies may be ineffective, resulting in large and irreducible interoceptive prediction errors. These gINS low-order

interoceptive prediction errors are then passed forward and recruit the metaceptive functions of the immediate insular module in the hierarchy, the dINS. The IMAC model hypothesizes that the dINS, as part of the DLPFC-mStriatum loop, is specialized for model-based interoceptive predictions of visceral or physiological states important for future action: cf., an allostatic mechanism to resolve interoceptive prediction errors at lower levels, which cannot be resolved through homeostatic reflexes. Furthermore, dINS model-based interoceptive predictions may initiate allostatic responses that adjust the functioning of visceral systems even before future actions or actual physiological disturbances occur [7,34]. Under model-based behaviour, dINS interoceptive predictions are selected by dopaminergic signals from the medial SN complex onto the mStriatum. A model-based strategy, however, has its limitations, given that it may be time inefficient, due to high computational cost and time constraints, to predict the consequences of all hypothetical actions and environmental state transitions. Conversely, a crucial advantage of a model-based strategy is that it can significantly accelerate learning of novel behaviours [150,153]. Regarding visceral systems, optimizing the learning of interoceptive representations by model-based adaptive behaviour could be vital for acquisition and maintenance of visceral responses that underwrite survival.

Model-based interoception may also contribute to realization of certain interoceptive states when individuals replay past experiences, imagine hypothetical situations and empathize with others. For example, the insular cortex seems to possess mirror-neuron-like functions that support empathetic behaviour and understanding of others' feelings, which are then associated with physiological responses, such as crying when observing others crying, crying in grief, imagery of one's own and others' body sensations and yawning contagiousness [181–185]. Further support for this view comes from neuroimaging studies showing insula activity associated with music-induced feelings, art aesthetic judgement, interoceptive imagery and retrieval of highly arousal, aversive, danger or disgusting experiences that have induced physiological changes [186–190].

Humans and other animals constantly face novel situations that require learning of entirely novel behaviors, especially when old behaviors prove suboptimal and inefficient, or when internal models are anachronistic, outdated and unreliable. Recent evidence suggests that exploratory behaviour, implemented in the VMPFC-vStriatum network, can support adaptive behaviour in situations in which the SMA-pStriatum and DLPFC-mStriatum fail to generate optimal behavioural policies [150,153,191,192]. The IMAC model hypothesizes that the metaceptive function of the aINS, as part of the VMPFC-vStriatum network, is specialized for interoceptive predictions that support concurrent mapping of visceral responses to novel actions that proved successful during exploratory behaviour. When allostatic interoceptive predictions by the dINS fail, its interoceptive prediction error signals, modulated by dopaminergic precision signals in the medial SN complex, are sent forward to the aINS, which is then recruited to engage exploratory behaviour to support acquisition of novel interoceptive policies. Under exploratory behaviour, aINS interoceptive predictions are selected by dopaminergic signals from the VTA synapsing onto the vStriatum.

Accumulating evidence supports several aspects of the IMAC model (it is not our goal here to provide a full review of the literature on studies that support the model). For instance, a meta-analysis of human neuroimaging studies showed that sensorimotor and visceral tasks activate the mid and posterior insula, cognitive tasks activate the dorsal anterior insula, and emotional tasks activate the ventral anterior insula [100]. Human and animal studies using tasks with visual or auditory cues in which participants have to apply knowledge of cue-outcome contingencies to predict future consequences of cues and actions show higher activity in the anterior insula, DLPFC, VMPFC and anterior striatum, with some studies also reporting anticipatory physiological responses associated with anticipatory neural activity [103,134,135,140,153,193–203]. By contrast, tasks that deliver unpredictable interoceptive stimulation found higher activity in the middle and posterior insula [183,184,198,199,204–208]. Furthermore, while imagination of sensory touch is linked with activity in the anterior insula and DLPFC, actual sensation of sensory touch is linked with activity in the posterior insula and primary somatosensory and motor cortices [183,184]. These findings are in general agreement with predictions made by the IMAC model that the anterior insula, including the dINS and aINS, are recruited in during behaviors that required planning and novel exploratory learning situations, whereas the gINS is recruited for processing information arriving directly from visceral systems.

Adaptive motor behaviour goes through stages of learning and transitions of control in the brain, with early learning, e.g. exploratory and model-based stages, under deliberative control and late learning under habitual and automatic control [129,130,148–150,152,209]. Similarly, the IMAC model hypothesizes that the aINS and dINS furnish interoceptive predictions in early stages of learning that

require exploratory or model-based interoceptive strategies, respectively, whereas the posterior gINS is recruited for interoceptive predictions once visceral responses have become habitual after repeated realizations. However, there are situations in which learning by exploratory behaviour is needed, but may eventually require consolidation of simple and quickly learned interoceptive predictions. In such situations, it may be inefficient and life-threatening to transit to a more complex and time-consuming dINS, model-based interoceptive strategy. The IMAC model hypothesizes that direct anatomical input from the aINS onto the gINS may serve as an efficient neural pathway for fast learning and consolidation of novel but possibly simple interoceptive representations, such as in Pavlovian conditioning.

Other neural systems also contribute to acquisition of interoceptive representations, given their prominent direct or indirect connections with the insular cortex. For instance, the TPJ, the amygdala, hippocampus and anterior cerebellum are well known for their participation in social cognition, classical conditioning, episodic and declarative memory and encapsulation of sensory-motor mappings, respectively [147,210–212]. The IMAC model hypothesizes that the TPJ may contribute to formation of social interoceptive predictions, e.g. visceral or physiological response in social relations [213]. The hippocampus may contribute to acquisition of declarative and episodic interoceptive predictions [214]. The amygdala may contribute to formation of Pavlovian interoceptive predictions [215], and the anterior cerebellum, which has neuroanatomical loop connections with the SMA [216,217] and is implicated in sequential learning and acquisition of forward models [129,147,150], may also be able to generate encapsulated interoceptive predictions to produce or coordinate automatic sequential visceral responses [129,147,212,218].

The insula active interoceptive inference neural architecture and hypotheses put forward above address several missing issues left unexplained in previous models of insula function. For instance, previous models using predictive and error-correction approaches [14,25] or information integration [20,21] sought to suggest specialized functions for the insula, based on how its local architecture processes a multitude of visceral, cognitive and emotional inputs it receives and its activation across multiple task domains. By contrast, the IMAC model assigns interoceptive inference functions to the insula based supported by the parallel network connections it forms with the PFC and striatum and their well-known roles in adaptive behaviour as well as the neuromodulatory input from dopaminergic and acetylcholinergic systems onto these networks. Thus, the IMAC framework can explain why neural activity of the insula is found in various emotional, motivational, social and cognitive tasks. Another difference in relation to previous models is our three-layer hierarchical architecture with first-order, second-order, and third-order interoceptive representations located in brainstem–subcortical systems, insular cortex, and PFC, respectively. Here, each hierarchical level is defined based on its intrinsic functional properties to generate innate autonomic reflexes or first-order interoceptive predictions, e.g. the brainstem and subcortical regions, or more flexible higher-order interoceptive predictions in the insula, e.g. second-order, and PFC, e.g. third-order. However, the organization of neural pathways connecting the ANS with the brain is more complex than a simple functional three-layered model (figure 2) and has recently been recognized in the neuroanatomical eight-layer hierarchical neurovisceral integration (NVI) model [219]. Despite different numbers of hierarchical layers, the IMAC and NVI models apply the same principles of predictive coding and Bayesian belief updating to suggest how interoceptive representations emerge at each hierarchical level. Future models or updated versions of both IMAC and NIV models should consider in more detail how to define levels of hierarchical interoceptive organization based on the number of synaptic connections linking the visceral systems to the insular cortex, their innate or flexible interoceptive representations, the interaction between the parallel interoceptive pathways (figure 3c), as well as the local cellular and molecular circuitry of the visceral systems, e.g. cardiac pace maker cells or the direct influences of hormones and other circulating chemicals on visceral functions.

# 4. Roles of dopamine and acetylcholine in interoceptive inference

Acetylcholine increases synaptic transmission in the thalamus, hippocampus and PFC, making the activity of neurons in these regions more responsive to synaptic input from other brain areas, and facilitating experience-dependent learning [220–222]. The IMAC model hypothesizes that the higher density of acetylcholine receptors in the aINS and dINS endows these regions with greater capability to flexibly modify and update prior metaceptions or learn novel metaceptions. Visceral responses can be triggered by stimulation of either the posterior or anterior insula, although stimulation and lesions

**Figure 3.** (*a*) Common view of cortical interoceptive control. In this view, the insular cortex, especially its agranular region, together with limbic structures, e.g. amygdala and VMPFC, receive ascending visceral signals and send descending control signals to modulate the activity of visceral systems. (*b*) The IMAC model proposes a hierarchical architecture where forward connections (green arrows) transmit interoceptive prediction errors, computed at each level of the hierarchy, and backward connections (red arrows) generate interoceptive predictions of desired interoceptive states. In this hierarchical interoceptive architecture, deep brain systems (subcortical and brainstem areas) contain mesaceptions (autonomic reflex arcs) or 1st order representations of interoceptive predictions. The insular cortex contains metaceptions or 2nd order representations of mesaceptions, and the PFC contains 3rd order representations of insula metaceptions and implements introspection to guide adaptive interoceptive behaviors and generate the meaning and causes of interoceptive prediction errors, as well as the consequences of interoceptive prediction. Gradient colors represent hierarchy levels within the insula, PFC and striatum (green: low hierarchy; orange: intermediate hierarchy; red: high hierarchy). (*c*) Examples of parallel insula hierarchical interoceptive networks. Here, five networks and their putative roles are highlighted with interoceptive signals arriving from either the vagus nerve (VN) and area postrema (AP) or from the spinothalamic tract (STT). Abbreviations: NTS (nucleus tractus solitary), MRF (medullar and reticular formation), PBN (parabrachial nucleus), PAG (periaqueductal gray), HYP (hypothalamus), THA (thalamus), gINS (granular insula), aINS (agranular insula), VTA-SNc (dopaminergic ventral tegmental area and substantia nigra pars compacta), DR-MR (serotonergic dorsal and median raphe nuclei), Amyg (amygdala), Hb (habenula), Str (striatum). Green arrows represent ascending interoceptive prediction errors and red arrows represent descending interoceptive predictions.

of the posterior and mid insula can exert wider influences on visceral processes, such as modulation of heart rate, blood pressure, kidney, bladder and gastric functions, and deficits in pain and thermal sensations [93,204,223–230]. The IMAC model suggests that the lower density of acetylcholine receptors in the posterior gINS serves to maintain the stability of prior interoceptive predictions that directly impact visceral survival functions, whereas the higher density of acetylcholine receptors in the anterior INS supports the flexible learning of novel interoceptive predictions. In short, the neuroplastic potential in the interoceptive (insular) hierarchy increases with hierarchical depth, enabling the learning of deep generative models of the embodied self.

The striatum has strong bidirectional connections with the dopaminergic system [81,231]. The function of dopaminergic neurons is associated with value-based learning and signalling rewards, aversive cues and alerting signals [155,232,233]. Striatal direct and indirect pathways have been implicated in facilitation versus inhibition of actions [234,235], learning good versus bad values [236] and reward versus aversive learning, respectively [237]. Recent evidence, however, suggests that striatal direct and indirect pathways do not employ on-off activation, but rather concurrent control of precise desirable and imprecise undesirable actions [238,239]. In the context of active inference, the striatum is responsible for selection of cortical representations based on precision signals mediated by dopaminergic neurons [30]. Based on the above findings, the IMAC model hypothesizes that dopaminergic input to the striatum also signals the precision of interoceptive predictions arriving from the insular cortex, by selectively increasing postsynaptic sensitivity to insular afferents. (Please

see [240] for a simulation of this dopaminergic selection, in the context of predictive coding and hierarchical motor control in Parkinson's disease.) The exact role of striatum direct and indirect pathways on interoceptive processes has yet to be determined. One possibility is that striatum direct and indirect pathways are concurrently engaged in learning and selection of interoceptive representations that promote survival and suppression of interoceptive representations that may cause harm, respectively.

At the cellular level, the IMAC model provided above a mechanistic, but simplistic interpretation of how striatum direct and indirect pathway medium spiny neurons and their dopaminergic input may compute interoceptive confidence signals, i.e. precision, of insula descending interoceptive predictions (figure 2), in much the same way as it computes confidence about actions and decisions represented in the PFC-striatum network [29,30,152,155,165,166,236,240,241]. By contrast, another model has suggested a mechanism for estimation of interoceptive confidence implemented by precision units located within the laminar structure of the insula [14]. Future work needs to establish the exact differences in the confidence signals computed at the insula laminar structure and the insula-striatum-dopamine network. The dopaminergic system also has direct projections to the insula (figure 3c) [231,242,243]. Precise roles of dopaminergic-insula connectivity have yet to be established, such as whether this pathway modulates synaptic plasticity or postsynaptic sensitivity involved in cognitive or movement control hypothesized for the dopaminergic-PFC pathway [240,244,245]. Similarly, we suggested that acetylcholinergic input onto the insular cortex may facilitate information transmission from other cortical regions, such as proposed for the PFC, hippocampus and thalamus [222]; however, the striatum also receives direct input from the cholinergic system, but the role of cholinergic insula and cholinergic striatum projections have yet to be established [246–248].

# 5. Insula and representations of conscious feelings

In order to simplify our treatment, we will use here a general notion found in the literature that emotions are unconscious arousal states linked with visceral and physiological processes under reflexive control, and feelings are conscious representations of emotions [14,21,23,249–257]. Subcortical and brainstem systems store neural representations of unlearned motivational drives that are capable of generating innate behavioural repertoires, e.g. consummatory, freezing, approach, avoidance and aggression [3,4,210,258,259]. Interestingly, these innate behavioural repertoires are also associated with background automatic visceral responses, e.g. increased heart rate, high blood pressure, bowel evacuation, pupil dilation, salivation and sweating, that are thought to elicit basic innate emotions, such as fear, anger, hunger, disgust, happiness, pleasure and surprise [210,249,251,260,261]. These findings led to influential neuropsychological theories proposing that emotions arise from a combination of interoceptive signals triggered by physiological changes in the functioning of visceral systems and associated behavioural repertoires [210,249,250,256]. The IMAC model suggests that mesaceptions, interoceptive predictions and interoceptive prediction errors computed in subcortical brain regions, e.g. amygdala, and in brainstem nuclei, e.g. NTS, reticular formation nuclei, PAG, PBN, dopaminergic and serotonergic nuclei, give rise to basic emotions or emotional substrata. For example, low glucose and insulin afferent signals onto brainstem systems generate interoceptive prediction errors that activate a mesaceptive representation of the emotion of hunger and trigger homeostatic responses and food-specific consumption behaviours through a specialized neural pathway in the hypothalamus [259,262]. This interpretation that emotions arise from brainstem interoceptive prediction errors is consistent with previous proposals that emotions emerge from dynamics in the rate of change, increase or decrease, of free-energy or interoceptive prediction errors triggered by visceral and physiological deviations from their expected functional parameters or setpoints [29]. The hypothalamus and other subcortical and brainstem systems, e.g. NTS, medullary reticular formation, area postrema, PBN, PAG and amygdala, have specialized and interacting neural survival circuits involved in interoceptive processing that may give rise to other mesaceptions involved not only in emotions related to feeding but also in emotions related to drinking, sex, aggression, excitement, fear, thermoregulation and neural immunity [210,259,263,264].

There have been multiple suggestions that cortical brain regions, including the insular cortex, contribute to generation of conscious feelings emerging from emotions [21,22,249,265–270]. Involvement of the insula in conscious feelings is suggested on the basis of the convergence of multiple types of somatic, visceral, motor, environmental, emotional, motivational, social and cognitive signals onto its structure [21,22,257]. However, this view is not immediately supported by

studies showing that patients with bilateral insula lesions are still able to consciously report feelings and self-awareness [271–275]. Although we generally agree with the idea that the insula may participate in consciousness processes, we believe that the insula alone may not be capable of generating consciousness of feelings and bodily states.

Then how does consciousness of feelings emerge from insula metaception? The IMAC model offers a specific hypothesis that consciousness of feelings and bodily states emerges from insula metaceptive representations and insula-PFC interactions, and is built up from experiences, innate emotional states, visceral and physiological responses associated with them. This hypothesis is informed by findings showing that the hierarchical organization of interoceptive representations mapped onto the modular architecture of the insula resembles the hierarchical organization of cognitive processes mapped onto the PFC and striatum [70,276]. Furthermore, human functional neuroimaging and stimulation studies have implicated functions of the SMA, DLPFC and VMPFC in conscious processes [268,277–282], modulation of visceral and physiological responses [283–285] and cognitive processes, such as automaticity, introspection, reasoning, categorization, interpretation of meanings and concept formation [129,286–291]. The pattern of insula-PFC anatomical connectivity allows the PFC to form higher- or third-order interoceptive representations, and to use complex cognitive functions, such as introspection, to inspect, interpret, categorize and reason on the contents, causes, and consequences of second-order interoceptive representations furnished by the insular cortex. For instance, in a situation in which someone experiences an aversive event that leads to an increase in heart rate, the insula will send interoceptive prediction errors to the PFC, which may interpret, categorize and contextualize them as a speeding heart associated with an imminent threat or a simple change of body posture, or after having a meal, the PFC may signal whether uncomfortable stomach signals indicate an unpleasant meal or an overly distended stomach caused by overeating. The insula-PFC networks may also support generalization of basic emotions into more complex feelings, such as social fear and anxiety, embarrassment, pride, guilt and grief. On a constructivist reading, these metaceptive representations provide the best explanation for the constellation of interoceptive, exteroceptive and proprioceptive representations at lower levels, furnishing predictions that resolve lower-level prediction errors.

Empirical findings, showing PFC-striatum networks engaged in distinct stages of learning, e.g. the DLPFC-mStr and VMPFC-vStr in early flexible-conscious learning and the SMA-pStr in late habitual-unconscious learning, suggest that even among insula-PFC parallel networks there may be hierarchical representations of conscious feeling. Accordingly, the lower-order gINS-SMA network may generate implicit or habitual metaceptions commonly associated with 'gut' or intuitive feelings that may be subpersonal and may be acquired after repeated experiences and may support the implementation of habitual action-interoception policies in response, for instance, to emotionally salient events, e.g. quickly escape from a snake attack or protect oneself from a sport injury. By contrast, the higher-order dINS-DLPFC and aINS-VMPFC networks may contribute to representations of explicit, introspective, conscious feelings that contribute to understanding one's physiological and visceral states in the early stages of learning of novel experiences or to reason and implement action-interoception policies, e.g. behavioural decisions, visceral and physiological responses, that solve current or future demands of the body and environment, e.g. cooking a meal in anticipation of high hunger or turning on the air conditioner to decrease indoor and body temperature. These higher-order conscious feelings may also be used to generate action-interoception policies that solve lower-order interoceptive representations and associated interoceptive prediction errors arriving from the gINS-SMA network or from the brainstem; i.e. affective qualia that contextualize belief updating at lower levels. According to this view, lower-order implicit or habitual metaceptions of the gINS-SMA network and mesaceptions may be assimilated into conscious processing, as they inform belief updating in insula-PFC networks.

Here, we proposed a mechanism of how insula-PFC networks may be involved in the generation of consciousness of feelings and bodily states. Other works have suggested, however, that conscious feelings emerge from other cortico-cortical, cortical-brainstem pathways or within the brainstem and hypothalamus [42,167,271,273,292]. The exact roles and the nature of interactions and contributions of these neural pathways for conscious feelings have yet to be identified.

# 6. IMAC implications for understanding depression

Patients suffering from depressive disorder exhibit multiple somatic and visceral symptoms, such as fatigue, reduced libido, disturbances in heart rate, blood pressure, gastrointestinal function, pupil

dilation, skin conductance, neuro-immunological dysfunctions and metabolic syndromes [293–298]. Somatic and visceral symptoms also predict the development and persistence of depression [299–302]. Associated with somatic symptoms are structural and functional abnormalities of the insular cortex and other cortical and subcortical brain regions involved in interoception [303–315]. Issues such as how visceral disturbances come about, how they change from an acute state to a chronic state with an almost irreversible point-of-no-return in treatment-resistant depressive patients, and what neural mechanisms are recruited to defend visceral systems, reset their healthy functions and protect humans from developing mood disorders, have yet to be clarified. Previous works have already described how disturbances in interoceptive Bayesian belief updating, e.g. of priors or internal representations, predictions and precision, may explain depression and other psychiatric disorders [14,23,24,36,316]. Below, we consider how the IMAC neural architecture and the hierarchical organization of the insula interoceptive pathways may prevent acute aversive experiences from developing into severe depression.

Interoceptive representations located in the brainstem, brainstem-gINS, brainstem-subcortical and insula-PFC-striatum networks can be used, independently or in combination, as neural defense systems against aversive and stressful experiences that may have negative impacts upon mental health and functioning of visceral systems. One important feature of the first-order interoceptive representations or mesaceptions, located in brainstem and subcortical regions, is their involvement not only in generation of neural activity necessary to maintain the body's survival functions, but also in control of viscero-chemical processes, e.g. hypothalamic-pituitary-adrenal axis, corticotropin-releasing hormone, energy metabolism, that are dysfunctional in depressive states and have detrimental consequences on the body, such as accumulation of visceral adipose, insulin resistance, cardiovascular problems, suppression of thyroid and reproductive functions and release of cortisol that have toxic effects on brain synaptic plasticity, structure and function [317–320].

Aversive experiences that are processed unconsciously in the brain or that may be perceived as non-threatening, but still generate atypical visceral responses, e.g. increased heart rate or increased cortisol, and interoceptive prediction errors, may be resolved at the brainstem level, e.g. NTS, MRF, PBN, PAG, by use of mesaceptions and implementation of innate interoceptive policies or higher-order habitual interoceptive policies located in the gINS-SMA network that can quickly correct interoceptive prediction errors and their ensuing visceral and viscero-chemical responses, thereby precluding permanent harmful physiological changes [321]. Candidate brainstem–subcortical systems include regions implicated in supervised, Pavlovian and instrumental learning of aversive and rewarding behaviors such as the cerebellum, amygdala, habenula and striatum–dopaminergic system [210,232,233,310] (figure 3c). However, highly salient aversive experiences may generate impactful interoceptive prediction errors that first-order or habitual interoceptive representations may be unable to resolve. In such situations, the body needs to rely on more elaborate cognitive processes, e.g. attention, working memory, introspection and interoceptive strategies, e.g. model-based, exploratory, implemented in the insula-PFC-striatum networks. Thus, while brainstem visceral nuclei may serve as the first line of defense against aversive experiences, and for the preservation of mental health, the brainstem-gINS, brainstem-subcortical and insula-PFC-striatum networks can serve as extended lines of defense, given that their learning functions underwrite the formation of novel, flexible and stereotypical interoceptive representations.

The extension of the hypothetical functions proposed in the IMAC architecture, to a defense system against depression, may be compromised when a pathological depressive state is present and linked with significant physiological, visceral and brain abnormalities. For instance, research with depressive patients in our laboratory has revealed four main findings [unpublished]: (i) abnormal heart rate variability, including increased heart rate and diminished sympathetic and parasympathetic control; (ii) bilateral reduced structural volume of brain regions involved in interoception, including the aINS, dINS, gINS, ACC, amygdala and hypothalamus; (iii) volume abnormalities of these regions associated with the degree of depressive symptoms, e.g. mood, somatic and visceral; (iv) reduced volume of the gINS linked to higher disturbances of sympathetic origin. These findings demonstrate the existence of widespread structural abnormalities in first-order, e.g. amygdala and hypothalamus, and second-order (insula) interoceptive systems generating Pavlovian interoceptive predictions as well as higher-order interoceptive predictions (habitual, model-based or exploratory) for cardiac control and possibly other visceral disturbances observed in depression. There are pharmacological and cognitive mechanisms, not explored in the IMAC model, that appear to reduce depressive symptoms and reverse visceral disturbances, such as cognitive behavioural therapy, drug treatments and neurofeedback [298,322–325].

# 7. Conclusion

In this opinion piece, we applied the active inference framework to propose the IMAC model in an attempt to explain how the hierarchical modular organization of the insular cortex supports formation of higher-order cortical interoceptive representations. The IMAC framework contrasts with the view that interoceptive afferents simply impress themselves on the central nervous system (figure 3a). Contributions of the IMAC model can be summarized in four main points (figure 3b): (i) parallel networks linking the insular cortex with the PFC and striatum are specialized for hierarchical generation of interoceptive policies that map interoceptive predictions to particular behaviors in an experience-dependent manner; (ii) the dopaminergic system emits precision signals quantifying the confidence of the insular interoceptive predictions; acetylcholine is hypothesized to underwrite plasticity in formation of novel interoceptive mappings by the dINS and aINS, and stability of interoceptive predictions in the gINS implicated in maintenance of visceral survival functions; (iii) two novel concepts, metaception and mesaception, were introduced to distinguish interoceptive representations in the insular and brainstem-subcortical systems, respectively; (iv) mesaceptions and metaceptions can explain emergence of unconscious emotions, conscious feelings and the rise of visceral dysfunctions observed in depression.

Future work using neuroimaging methods with humans is needed to test predictions made by the IMAC model, such as the dissociation of involvement of the parallel insula–PFC–striatum networks in distinct stages of formation of interoceptive representations and in the hierarchical representation of conscious feelings. Anatomical and functional neuroimaging studies also need to identify not only how disturbances in cortical, subcortical and brainstem systems contribute to development of visceral dysfunctions and mood disorders but also how such disturbances come about, such as whether neural degeneration of the insular cortex starts in its agranular region and progresses to its granular region, or the other way around. The field of interoception neuroimaging is progressing steadily with human studies, but it will also benefit from experimental studies with animals, to provide a more detailed map of the molecular, cellular and neuroanatomical connections among visceral organs and how their interoceptive signals are interpreted by the brain.

Data accessibility. This article has no additional data.

Authors' contributions. A.SR.F.: conceptualization, funding acquisition, writing—original draft, writing—review and editing; K.J.F.: conceptualization, funding acquisition, writing—original draft, writing—review and editing; S.Y.: conceptualization, funding acquisition, writing—original draft, writing—review and editing.

All authors gave final approval for publication and agreed to be held accountable for the work performed therein.

Conflict of interest declaration. The authors declare no competing interests.

Funding. A.F. and S.Y. are grateful for research support provided by the Japan Agency for Medical Research and Development (AMED), grant no. JP19dm0107093, the COI STREAM program from the Japan Science and Technology Agency (JST), grant no. JPMJCE1311 and JPMJCA2208, the Ministry of Education, Culture, Sports, Science and Technology, grant no. 20309202, and the Japanese Society for the Promotion of Science, grant no. 20K07723 and Moonshot-9 (MS9) JPMJMS2296. K.F. is supported by funding from the Wellcome Centre for Human Neuroimaging (ref: 205103/Z/16/Z) and a Canada-UK Artificial Intelligence Initiative (ref: ES/T01279X/1).

Acknowledgements. We are grateful to the reviewers for their insightful comments.

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
