## [Peer Review File · Royal Society Open Science]

Review History

RSOS-201477.R0 (Original submission)

Review form: Reviewer 1

Is the manuscript scientifically sound in its present form?

Yes

Are the interpretations and conclusions justified by the results?

Yes

Is the language acceptable?

Yes

Do you have any ethical concerns with this paper?

No

Have you any concerns about statistical analyses in this paper?

No

Recommendation?

Major revision is needed (please make suggestions in comments)

Comments to the Author(s)

General / abstract

The notion that insula is organized into hierarchical modules needs to be clarified

1) what is evidence that these are modules i.e. serving as discrete functional units

2) what is the evidence that they are hierarchical

To address this requires a deep analysis of the functional architecture of insular cortex

Several observations support modularity e.g. topographical segregation of taste cortex or inputs from heart versus gut suggest functional modularity but such observations may also work against notion of hierarchical organization I.e, not all representations start in project granular regions and project sequentially to agranular insula.

Sequential projection within insula for posterior to anterior is central to Craig's model of insular interoceptive representation and ultimately conscious feelings, and this was informed by his own anatomical studies. Similarly, neuroanatomical work of Helen Barbas has informed Barratt and colleagues own model of insula in interoception predictive representation.

Henry Evrard has undertaken the most comprehensive cytoarchitectonic studies of primate insula, identifying at least 22 subregions. And mapping inputs and outputs to each region. This should provide the anatomical context for understanding not only how insular organization fits the notion of an interoceptive hierarchy proposed in this paper.

The introduction jumps straight into a problem area: The distinction between efferent and afferent innervation of internal organs. Simply, the autonomic nervous system (ANS) is typically viewed as peripheral and efferent (sympathetic and parasympathetic nerves). This is distinct from interoception where viscerosensory afferent nerves carry interoceptive information centrally from visceral interoceptors. Interoceptive pathways may share nerves with autonomic efferents but not always. They also clearly do inform "autonomic" control (as do other channels of sensory information). However, if the authors want to conflate afferent interoception with efferent autonomic control they need to make a stronger case for doing so. Line 6-7 effectively says 'the motor system is responsible for transmitting somatosensory information from brain to body... this may be true at a stretch, in active inference terms, but the distinct afferent pathway mediating this is missing. Extending this to say the ANS is responsible for thermoregulatory hormonal and immune responses is a further stretch give humoral sampling by paraventricular organs is a different type of channel from central neural signalling from the periphery. TL:DR I take issue with the broad redefinition of the autonomic nervous system and its conflation with interoceptive signalling.

The second paragraph of the introduction sets up a straw man: the premise that insula is the main target for interoceptive afferents: it is certainly a target for projection from a thalamic area that receives interoceptive information, but one might argue brainstem nuclei notably the NTS, are the main target of interoceptive afferents. That aside, in this opinion piece, perhaps there needs to be an early, stronger definition of interoception and what the authors think it is for (i.e. what is the question the new model is trying to address). Statements like 'interoceptive adaptive behavior' can then be better understood.

Figure 1 b is puzzling: Leaving aside my view that it doesn't make functional sense to speak of parasympathetic and sympathetic afferents (I presume the authors are referring to viscerosensory afferents to brain respectively via vagus / glossopharyngeal nerve and spinal cord – the later encompassing presumably sacral parasympathetic afferents), the authors appear to be proposing that there is a direct (non-synapsing) neural connection from some viscera to granular insula and that all NTS information to thalamus relays via parabrachial nucleus hypothalamus where and PAG (really). Obligatory pontine relays from NTS are a feature of rodents but not primates.

The description of sympathetic and parasympathetic (efferent) pathways in the periphery is perhaps not central to the proposed model, but the statement that postganglionic sympathetic innervation of smooth muscle is cholinergic (P4 line 17,18) is wrong. Noradrenaline is the postganglionic neurotransmitter for all end organs with some few exceptions, notably sudomotor glands in adult humans. Circulating adrenaline thus also acts on the majority of sympathetically innervated organs. The parasympathetic innervation of the pelvis is not addressed for completion.

The statement “most visceral information , reaches the brain via the vagus nerve” P4 line 31/32) needs some kind of quantitative evidence i.e. a primary source, as I would dispute this based on observation from vagotomy (which, as an aside, has also been used in management of visceral pain P5 l2)

P5 l8-10 this description of central pathways to thalamus “before arriving at the insular cortex” is superficial; Craig has provided more detail about this e.g. how representations are mapped into different regions of thalamus, and he also identifies other thalamocortical projections (e.g. to cingulate of this information.

P5 l13-24 insula heirachical organization: This section is quite critical to the model and should draw from Everard’s work rather than the older papers cited. I am doubtful if virus tracing is the basis to definition of 3 larger cytoarchitectonic regions within insula, but may inform connections between regions. As noted above there is ore up to date information in primate brain about this. The arguments about cortical connectivity as a hierarchy, typically refers to input (e.g. granule cell) and output layers of cortex – what is the evidence here for hierarchy. “Reciprocal connections between dINS and gINS suggest no hierarchy between these two regions, is this correct?

P5 l27 is a bit confusing in that it suggests a flow of information, when what it describes is the location of e.g. primary taste cortex in monkey located in anterior (agranular insula) and thermal representations in granular posterior insula. This description of regional modularity could be made clearer. The beginning paragraph on page 6 is similarly selective: primary taste cortex in anterior insular and primary olfactory cortex in pre-pyriform and primary visual cortex do not project to gINS.

P5 The insular dopaminergic connection needs to be better described i.e. what cells do they innervate, and what is their function? In rodents agranular insula has a dopamine input (like OFC), with some identified functions. Dopamine may be especially relevant to the proposed IMAC model, but other neuromodulator’s (e.g. noradrenaline from locus ceruleus) have been implicated in similar models. A case needs to be made for selecting dopamine among for comment.

IMAC Model P8 – the first paragraph could be more explicit in associating active inference with) action or internal efferent responses to differentiate active inference within general predictive coding models. This may mean a clearer definition of policies (line 2).

What is meant by “innate” in line 8?

Lines 17-30 Is this true? Is interoceptive information (e.g. concerning respiration) represented in gINS represented in dINS and again in aINS (what is the evidence?). This model of insular information flow is similar to the proposed bottom-up model of Craig’s. And, is the cortical connectivity in insula specialized, i.e. different from elsewhere in the brain, for predictive coding? It there empirical evidence for interoceptive prediction error computations within dINS?

Figure 2 links articulates the IAC model with the proposal of parallel hierarchies for “interoceptive habit, model-based and exploration”. Again there is assumption that the selected

interregional connectivity supports hierarchical predictive processing: It is noteworthy how many brain regions are included in the model.

The IMAC model as described in the text is insightful. The point regarding similarity to known prefrontal striatal “loops” are well made, and the narrative how these parallel systems might work together is compelling and overcomes many limitations of earlier models that tend to over-generalize. The notion of model-based strategy formation for behavioral policy selection is intriguing – an example of how this works would be helpful (to ensure understanding of exactly how this might work – this seems relevant to earlier point about what is meant by “interoceptive adaptive behavior” as this could encompass many things at multiple levels. Much of the model lends itself to experimental predictions about how insular might interact with different parts of striatum in different situations.

Could these predictions be made more explicit, e.g. in a table?

P10, The section on dopamine and acetylcholine presumably relates to ACH in cortex (rather than striatum). in contrast to dopamine the relevant neuroanatomy (NBMeynert?) is not described earlier in the paper, nor illustrated in figure 2.

It took two reads to understand how the proposal related to the generation and representation of feelings. The distinction between mesa and meta ception) does seem to cover this nicely – one assumes feelings are always present and changes draw attention to these at different functional levels. Again the model is elegant in what is proposed; it’s parsing of different feeling states could be applied to understanding, for example alexithymia or disorders of empathy. The abstract suggests that this is relevant to conditions like addiction depression, PTSD and schizophrenia. This is not picked up later unfortunately. A compelling case was made recently regarding the metacognitive interoceptive basis to fatigue and depression. Although every disorder cannot be mentioned an IMAC account of fatigue and depression would be useful. With anxiety also linked to interoceptive prediction, how does the clinical expression of symptoms and behaviour fit within the model?

Review form: Reviewer 2

Is the manuscript scientifically sound in its present form?

No

Are the interpretations and conclusions justified by the results?

No

Is the language acceptable?

Yes

Do you have any ethical concerns with this paper?

No

Have you any concerns about statistical analyses in this paper?

No

Recommendation?

Reject

Comments to the Author(s)

Overview

This paper offers a detailed theoretical account of how the lobular structure of the insula under an active inference framework may support interoception. This is stated in relation to numerous other brain phenomena in which interoceptive processes play a role, such as feeling states, emotions, and memory. While this paper serves to extend previous theories, the authors can better ground the proposed framework in pre-existing models of the hierarchical neuroanatomy serving neurovisceral integration, and provide a more comprehensive view of different structures involved in interoceptive processing and prediction. In addition, the authors can more thoroughly define certain terms and concepts, and more clearly state where aspects of the model lack experimental evidence while offering ways in which the predictions of this model could be directly tested.

Main suggestions

- The proposed insula lobular model of interoception (and specifically the “Body-brain interoceptive pathways” section beginning on page 3) can be better grounded in pre-existing theories of neurovisceral integration that outline the hierarchical, autonomic regulatory loops below the level of the insular cortex (See Smith et al 2017, “The hierarchical basis of neurovisceral integration”, *Neurosci Biobehav Rev*, DOI: 10.1016/j.neubiorev.2017.02.003 for an outline of the hierarchical functional anatomy). This can help establish the level at which the insula acts, as well as the nature of information it receives. The use of terms such as “information [passing]” (page 5, line 7) insinuates passive relaying of information, without adequate mention to the filtering, integration, or interpretation of information that occurs below the level of the insula, or that which is affected by descending modulatory input above the level of the insula. Finally, the predictive coding model states that prediction errors at these lower levels are what will propagate upwards, not just passive information relay of sensory information per se.
- Organization of paper can be adjusted so that the functions of parallel networks linked with modules of the insula are introduced at the same time as merely describing the anatomy (“insula-cortical connections” and “insula-striatum-dopaminergic connections” sections). This would add flow to the paper, and help draw clearer links between the proposed functions and the neuroanatomy underpinning them. For example, the insula-cortical connections are only minimally mentioned early on in the paper, and the role of the PFC in interoceptive introspection is not adequately introduced.
- The following important terms are used often but not explicitly defined, and thus warrant specific attention. The authors could consider having a separate glossary for these terms.
 - o Interoceptive inference: see Seth & Friston (2016), “Active interoceptive inference and the emotional brain”, *Philos Trans R Soc Lond B Biol Soc*, DOI: 10.1098/rstb.2016.0007
 - o Interoceptive responses & control: Is this referring to the efferent output, mapped to certain levels/patterns of afferent input, that acts to regulate the present or future sensory interoceptive states?
 - o Optimal policies, behavioural policies: policies could map the pattern of sensory input to the pattern of visceromotor output, not only to overt behaviour per se. Additionally, how is “optimal” defined in different scenarios? Is it in relation to the homeostatic set-points, learned interoceptive response, and desired interoceptive states for the agent?
 - o Higher-, intermediate-, vs. lower-order predictions: is the only difference between those the insular subregion from which they are generated? What relation does this distinction have with the mesa- (cortical) and meso- (subcortical) conceptions stated earlier?
 - o Interoceptive habit, model-based, versus exploration: What are the explicit differences between these three constructs, other than the insular sub-region that they are associated with? What is their relationship to the interoceptive memories and interoceptive learning listed later on in the paper? How do they differentially influence the other motor, cognitive, and affective functions in which the insula is proposed to play a role?
- Other models have slightly different depictions for the flow interoceptive information from and within subcortical to cortical structures (Fig. 3 in this paper).

- o See Figure 4 from Seth & Friston 2016, “Active interoceptive inference and the emotional brain” (Philos Trans R Soc Lond B Biol Sci, doi: 10.1098/rstb.2016.0007) who include other structures such as amygdala, ACC, and OFC but do not include lobular structure of the insula. These authors also do not have the distinction between sub-cortical “emotions” and cortical “feelings”.
- o See Figure 1 of Barrett & Simmons 2015, “Interoceptive predictions in the brain” (Nat Rev Neurosci, doi: 10.1038/nrn3950), who do include hierarchical organization insular subregions but with a slightly different organization, and with cell bodies of precision cells residing in the insula itself.
- o Additionally, the caption of this figure could be more clear in that (a) is not necessarily incorrect, but instead just more simplified than (b) which expands upon it.
 - There is a large focus through the paper on the autonomic nervous system and visceral interoceptive processing. However, there are other pathways of interoception not mentioned in the paper that can play a role in many of the interoceptive and regulatory processes, emotions, and feeling states that are highlighted. Examples of these pathways include central chemoreceptors and hormones carried in the blood to the central nervous system (see Berntson & Khalsa 2021, “Neural circuits of interoception”, Trends in Neuroscience, DOI: 10.1016/j.tins.2020.09.011)
 - Instead of just listing the three terms, the title of the paper could be more cohesive and descriptive of the proposed relationship between them. For example, “feeling representation” is not mentioned until nearly the end of the paper, yet it is stated in the title. Additionally, although the title of the model (IMAC) emphasizes the modular architecture of the insula, it then de-emphasizes the other brain structures that play a role in interoceptive control and the visceral regulation.
 - The authors could more clearly state which aspects of the model lack experimental evidence, while at the same time offering ways in which parts of this model could be directly tested using experimental tasks or manipulations. For example, how could researchers design studies to investigate the three specific levels of interoceptive processing in the subregions of the insula (interoceptive habit, model-based, vs. exploration, or mesa- vs. meso-conception)? Furthermore, how could dysfunction in these distinct processes be linked to what is known of different mental health disorders? This latter point will also help increase the clinical relevance of adopting such a model and its distinctions, and how this model could help better understand disease processes, improve treatments, and predict patient outcomes.

Minor suggestions

- “Insula-visceral and somatic maps” section is vague, and a clearer description of topographic maps of the different sensations (thermal, somatosensory, visceral, etc.) could be provided.
- o Recent work argues against a topographic representation for gustatory stimuli (see Avery 2021, “Against gustotopic representation in the human brain: there is no Cartesian restaurant”, Curr Opin Physiol, DOI: 10.1016/j.cophys.2021.01.005).
- o Does “topographic representation of emotional, motivational, cognitive, and visceromotor processing” (page 6, line 11) mean that different emotions (i.e. sad, angry) or motivations (i.e. hunger, thirst) are linked to spatially adjacent but non-overlapping locations on the insula?
- o Do “insula-visceral and insula-somatic maps” refer to a proposed generative map or model of the body that may be stored in the insula, or simply a topographic organization of the sensory information?
 - The concept of allostasis should be mentioned earlier on in the paper, when discussing “adaptive interoceptive inference” and the active inference framework (Figure 2)
 - Page 8 line 18: “neural” should be “neural structures”
 - Page 11 line 20: “effect” should be “affect”

Optional suggestions

- The authors consider efferent, regulatory output under the definition of interoception, but could consider commenting on the distinction between afferent sensory and efferent regulatory arms.
- The introductory paragraph to begin the paper reads more like a list, and never defines interoception.
- Other cortical areas in addition to the insula can be outlined more fully, in relation to the processes they contribute to, rather than kept in one separate section towards the end. For example, other areas such as the cingulate cortex may contribute to the generation of visceromotor commands that can occur as part of motor, affective, and cognitive networks for the regulation of internal bodily states.
- In Figure 3, the authors could consider adding the dopaminergic areas that modulate the precision of signals.

Decision letter (RSOS-201477.R0)

Dear Dr Fermin

The Editors assigned to your paper RSOS-201477 "Insula Interoception, Active Inference and Feeling Representation" have made a decision based on their reading of the paper and any comments received from reviewers.

Regrettably, in view of the reports received, the manuscript has been rejected in its current form. However, a new manuscript may be submitted which takes into consideration these comments.

We invite you to respond to the comments supplied below and prepare a resubmission of your manuscript. Below the referees' and Editors' comments (where applicable) we provide additional requirements. We provide guidance below to help you prepare your revision.

Please note that resubmitting your manuscript does not guarantee eventual acceptance, and we do not generally allow multiple rounds of revision and resubmission, so we urge you to make every effort to fully address all of the comments at this stage. If deemed necessary by the Editors, your manuscript will be sent back to one or more of the original reviewers for assessment. If the original reviewers are not available, we may invite new reviewers.

Please resubmit your revised manuscript and required files (see below) no later than 05-Oct-2021. Note: the ScholarOne system will 'lock' if resubmission is attempted on or after this deadline. If you do not think you will be able to meet this deadline, please contact the editorial office immediately.

Please note article processing charges apply to papers accepted for publication in Royal Society Open Science (<https://royalsocietypublishing.org/rsos/charges>). Charges will also apply to papers transferred to the journal from other Royal Society Publishing journals, as well as papers submitted as part of our collaboration with the Royal Society of Chemistry (<https://royalsocietypublishing.org/rsos/chemistry>). Fee waivers are available but must be requested when you submit your manuscript (<https://royalsocietypublishing.org/rsos/waivers>).

Thank you for submitting your manuscript to Royal Society Open Science and we look forward to receiving your resubmission. If you have any questions at all, please do not hesitate to get in touch.

on behalf of Dr Anastasia Christakou (Associate Editor) and Essi Viding (Subject Editor)
 openscience@royalsociety.org

Associate Editor Comments to Author (Dr Anastasia Christakou):

Associate Editor: 1

Comments to the Author:

Thank you for the opportunity to read your work. Please accept my personal apologies for the delay in assessing your manuscript.

I trust you will find the reviews provided detailed and constructive. You will also note that both reviewers have raised serious issues with aspects of the manuscript. I would like to highlight 3 areas that need close attention and quite considerable revision:

Firstly, careful attention needs to be given to definitions of non-unitary constructs, such as “interoceptive adaptive behaviour”, “introspection” (putatively implemented by a single brain area), and “emotions” versus “feelings”. Ambiguity in such terms causes some confusion about the level of descriptive granularity at which the model is to be taken to operate. Relatedly, terms such as “habit” and “optimality” have extensive histories of analysis and debate, but are used in places without contextualisation.

Secondly, and relatedly, more careful attention needs to be paid to the description of functional anatomy, drawing on extensive and detailed work in the field. It is telling that some of the most appealing features of the model relate to bringing interoceptive inferential processes in line with compelling existing models of action control based on functional anatomy, but at the same time there is extensive criticism of the neuroanatomical detail as described in the manuscript. This is particularly important in situations where the descriptions (e.g.: of putatively directional connectivity) on which the model appears to be based disagree with equivalent descriptions in other models.

Thirdly, it is important to describe more explicitly and in detail the utility of the model for the field – what does it do for our understanding that other models do not achieve, beyond bringing together two currently prominent areas (interoception and active inference). More detailed analysis is needed especially when the model draws on or contrasts previous models. Relatedly, I highlight a suggestion by one of the reviewers who asks that it is made explicit when empirical evidence for aspects of the model’s assumptions or predictions is not available, and how this gap might be addressed.

I appreciate that the reviews call for substantial work to be put into revising the manuscript. I hope you will find the reviewers’ comments helpful in taking it on, and that you will choose to resubmit to the journal. I look forward to reading the next iteration.

Reviewer comments to Author:

Reviewer: 1

Comments to the Author(s)

General / abstract

The notion that insula is organized into hierarchical modules needs to be clarified

!) what is evidence that these are modules i.e. serving as discrete functional units

2) what is the evidence that they are hierarchical

To address this requires a deep analysis of the functional architecture of insular cortex. Several observations support modularity e.g. topographical segregation of taste cortex or inputs from heart versus gut suggest functional modularity but such observations may also work against notion of hierarchical organization I.e, not all representations start in project granular regions and project sequentially to agranular insula.

Sequential projection within insula for posterior to anterior is central to Craig's model of insular interoceptive representation and ultimately conscious feelings, and this was informed by his own anatomical studies. Similarly, neuroanatomical work of Helen Barbas has informed Barratt and colleagues own model of insula in interoception predictive representation.

Henry Evrard has undertaken the most comprehensive cytoarchitectonic studies of primate insula, identifying at least 22 subregions. And mapping inputs and outputs to each region. This should provide the anatomical context for understanding not only how insular organization fits the notion of an interoceptive hierarchy proposed in this paper.

The introduction jumps straight into a problem area: The distinction between efferent and afferent innervation of internal organs. Simply, the autonomic nervous system (ANS) is typically viewed as peripheral and efferent (sympathetic and parasympathetic nerves). This is distinct from interoception where viscerosensory afferent nerves carry interoceptive information centrally from visceral interoceptors. Interoceptive pathways may share nerves with autonomic efferents but not always. They also clearly do inform "autonomic" control (as do other channels of sensory information). However, if the authors want to conflate afferent interoception with efferent autonomic control they need to make a stronger case for doing so. Line 6-7 effectively says 'the motor system is responsible for transmitting somatosensory information from brain to body... this may be true at a stretch, in active inference terms, but the distinct afferent pathway mediating this is missing. Extending this to say the ANS is responsible for thermoregulatory hormonal and immune responses is a further stretch give humoral sampling by paraventricular organs is a different type of channel from central neural signalling from the periphery. TL:DR I take issue with the broad redefinition of the autonomic nervous system and its conflation with interoceptive signalling.

The second paragraph of the introduction sets up a straw man: the premise that insula is the main target for interoceptive afferents: it is certainly a target for projection from a thalamic area that receives interoceptive information, but one might argue brainstem nuclei notably the NTS, are the main target of interoceptive afferents. That aside, in this opinion piece, perhaps there needs to be an early, stronger definition of interoception and what the authors think it is for (i.e. what is the question the new model is trying to address). Statements like 'interoceptive adaptive behavior' can then be better understood.

Figure 1 b is puzzling: Leaving aside my view that it doesn't make functional sense to speak of parasympathetic and sympathetic afferents (I presume the authors are referring to viscerosensory afferents to brain respectively via vagus /glossopharyngeal nerve and spinal cord - the later encompassing presumably sacral parasympathetic afferents), the authors appear to be proposing that there is a direct (non-synapsing) neural connection from some viscera to granular insula and that all NTS information to thalamus relays via parabrachial nucleus hypothalamus where and PAG (really). Obligatory pontine relays from NTS are a feature of rodents but not primates.

The description of sympathetic and parasympathetic (efferent) pathways in the periphery is perhaps not central to the proposed model, but the statement that postganglionic sympathetic innervation of smooth muscle is cholinergic (P4 line 17,18) is wrong. Noradrenaline is the postganglionic neurotransmitter for all end organs with some few exceptions, notably sudomotor glands in adult humans. Circulating adrenaline thus also acts on the majority of sympathetically innervated organs. The parasympathetic innervation of the pelvis is not addressed for completion.

The statement “most visceral information , reaches the brain via the vagus nerve” P4 line 31/32) needs some kind of quantitative evidence i.e. a primary source, as I would dispute this based on observation from vagotomy (which, as an aside, has also been used in management of visceral pain P5 l2)

P5 l8-10 this description of central pathways to thalamus “before arriving at the insular cortex” is superficial; Craig has provided more detail about this e.g. how representations are mapped into different regions of thalamus, and he also identifies other thalamocortical projections (e.g. to cingulate of this information.

P5 l13-24 insula heirachical organization: This section is quite critical to the model and should draw from Everard’s work rather than the older papers cited. I am doubtful if virus tracing is the basis to definition of 3 larger cytoarchitectonic regions within insula, but may inform connections between regions. As noted above there is ore up to date information in primate brain about this. The arguments about cortical connectivity as a hierarchy, typically refers to input (e.g. granule cell) and output layers of cortex – what is the evidence here for hierarchy. “Reciprocal connections between dINS and gINS suggest no hierarchy between these two regions, is this correct?

P5 l27 is a bit confusing in that it suggests a flow of information, when what it describes is the location of e.g. primary taste cortex in monkey located in anterior (agranular insula) and thermal representations in granular posterior insula. This description of regional modularity could be made clearer. The beginning paragraph on page 6 is similarly selective: primary taste cortex in anterior insular and primary olfactory cortex in pre-pyriform and primary visual cortex do not project to gINS.

P5 The insular dopaminergic connection needs to be better described i.e. what cells do they innervate, and what is their function? In rodents agranular insula has a dopamine input (like OFC), with some identified functions. Dopamine may be especially relevant to the proposed IMAC model, but other neuromodulator’s (e.g. noradrenaline from locus ceruleus) have been implicated in similar models. A case needs to be made for selecting dopamine among for comment.

IMAC Model P8 – the first paragraph could be more explicit in associating active inference with action or internal efferent responses to differentiate active inference within general predictive coding models. This may mean a clearer definition of policies (line 2).

What is meant by “innate” in line 8?

Lines 17-30 Is this true? Is interoceptive information (e.g. concerning respiration) represented in gINS represented in dINS and again in aINS (what is the evidence?). This model of insular information flow is similar to the proposed bottom-up model of Craig’s. And, is the cortical connectivity in insula specialized, i.e. different from elsewhere in the brain, for predictive coding? It there empirical evidence for interoceptive prediction error computations within dINS?

Figure 2 links articulates the IAC model with the proposal of parallel hierarchies for “interoceptive habit, model-based and exploration”. Again there is assumption that the selected interregional connectivity supports hierarchical predictive processing: It is noteworthy how many brain regions are included in the model.

The IMAC model as described in the text is insightful. The point regarding similarity to known prefrontal striatal “loops” are well made, and the narrative how these parallel systems might work together is compelling and overcomes many limitations of earlier models that tend to over-generalize. The notion of model-based strategy formation for behavioral policy selection is intriguing – an example of how this works would be helpful (to ensure understanding of exactly how this might works – this seems relevant to earlier point about what is meant by “interoceptive adaptive behavior” as this could encompass many things at multiple levels. Much of the model

lends itself to experimental predictions about how insular might interact with different parts of striatum in different situations.

Could these predictions be made more explicit, e.g. in a table ?

P10, The section on dopamine and acetylcholine presumably relates to ACH in cortex (rather than striatum). in contrast to dopamine the relevant neuroanatomy (NBMeynert?) is not described earlier in the paper, nor illustrated in figure 2.

It took two reads to understand how the proposal related to the generation and representation of feelings. The distinction between mesa and meta ception) does seem to cover this nicely – one assumes feelings are always present and changes draw attention to these at different functional levels Again the model is elegant in what is proposed; it's parsing of different feeling states could be applied to understanding, for example alexithymia or disorders of empathy. The abstract suggests that this is relevant to conditions like addiction depression, PTSD and schizophrenia. This is not picked up later unfortunately. A compelling case was made recently regarding the metacognitive interoceptive basis to fatigue and depression. Although every disorder cannot be mentioned an IMAC account of fatigue and depression would be useful. With anxiety also linked to interoceptive prediction, how does the clinical expression of symptoms and behaviour fits within the model?

Reviewer: 2

Comments to the Author(s)

Overview

This paper offers a detailed theoretical account of how the lobular structure of the insula under an active inference framework may support interoception. This is stated in relation to numerous other brain phenomena in which interoceptive processes play a role, such as feeling states, emotions, and memory. While this paper serves to extend previous theories, the authors can better ground the proposed framework in pre-existing models of the hierarchical neuroanatomy serving neurovisceral integration, and provide a more comprehensive view of different structures involved in interoceptive processing and prediction. In addition, the authors can more thoroughly define certain terms and concepts, and more clearly state where aspects of the model lack experimental evidence while offering ways in which the predictions of this model could be directly tested.

Main suggestions

- The proposed insula lobular model of interoception (and specifically the “Body-brain interoceptive pathways” section beginning on page 3) can be better grounded in pre-existing theories of neurovisceral integration that outline the hierarchical, autonomic regulatory loops below the level of the insular cortex (See Smith et al 2017, “The hierarchical basis of neurovisceral integration”, *Neurosci Biobehav Rev*, DOI: 10.1016/j.neubiorev.2017.02.003 for an outline of the hierarchical functional anatomy). This can help establish the level at which the insula acts, as well as the nature of information it receives. The use of terms such as “information [passing]” (page 5, line 7) insinuates passive relaying of information, without adequate mention to the filtering, integration, or interpretation of information that occurs below the level of the insula, or that which is affected by descending modulatory input above the level of the insula. Finally, the predictive coding model states that prediction errors at these lower levels are what will propagate upwards, not just passive information relay of sensory information per se.
- Organization of paper can be adjusted so that the functions of parallel networks linked with modules of the insula are introduced at the same time as merely describing the anatomy (“insula-cortical connections” and “insula-striatum-dopaminergic connections” sections). This would add flow to the paper, and help draw clearer links between the proposed functions and the neuroanatomy underpinning them. For example, the insula-cortical connections are only

minimally mentioned early on in the paper, and the role of the PFC in interoceptive introspection is not adequately introduced.

- The following important terms are used often but not explicitly defined, and thus warrant specific attention. The authors could consider having a separate glossary for these terms.
 - o Interoceptive inference: see Seth & Friston (2016), “Active interoceptive inference and the emotional brain”, *Philos Trans R Soc Lond B Biol Soc*, DOI: 10.1098/rstb.2016.0007
 - o Interoceptive responses & control: Is this referring to the efferent output, mapped to certain levels/patterns of afferent input, that acts to regulate the present or future sensory interoceptive states?
 - o Optimal policies, behavioural policies: policies could map the pattern of sensory input to the pattern of visceromotor output, not only to overt behaviour per se. Additionally, how is “optimal” defined in different scenarios? Is it in relation to the homeostatic set-points, learned interoceptive response, and desired interoceptive states for the agent?
 - o Higher-, intermediate-, vs. lower-order predictions: is the only difference between those the insular subregion from which they are generated? What relation does this distinction have with the mesa- (cortical) and meso- (subcortical) conceptions stated earlier?
 - o Interoceptive habit, model-based, versus exploration: What are the explicit differences between these three constructs, other than the insular sub-region that they are associated with? What is their relationship to the interoceptive memories and interoceptive learning listed later on in the paper? How do they differentially influence the other motor, cognitive, and affective functions in which the insula is proposed to play a role?
- Other models have slightly different depictions for the flow interoceptive information from and within subcortical to cortical structures (Fig. 3 in this paper).
 - o See Figure 4 from Seth & Friston 2016, “Active interoceptive inference and the emotional brain” (*Philos Trans R Soc Lond B Biol Sci*, doi: 10.1098/rstb.2016.0007) who include other structures such as amygdala, ACC, and OFC but do not include lobular structure of the insula. These authors also do not have the distinction between sub-cortical “emotions” and cortical “feelings”.
 - o See Figure 1 of Barrett & Simmons 2015, “Interoceptive predictions in the brain” (*Nat Rev Neurosci*, doi: 10.1038/nrn3950), who do include hierarchical organization insular subregions but with a slightly different organization, and with cell bodies of precision cells residing in the insula itself.
 - o Additionally, the caption of this figure could be more clear in that (a) is not necessarily incorrect, but instead just more simplified than (b) which expands upon it.
- There is a large focus through the paper on the autonomic nervous system and visceral interoceptive processing. However, there are other pathways of interoception not mentioned in the paper that can play a role in many of the interoceptive and regulatory processes, emotions, and feeling states that are highlighted. Examples of these pathways include central chemoreceptors and hormones carried in the blood to the central nervous system (see Berntson & Khalsa 2021, “Neural circuits of interoception”, *Trends in Neuroscience*, DOI: 10.1016/j.tins.2020.09.011)
- Instead of just listing the three terms, the title of the paper could be more cohesive and descriptive of the proposed relationship between them. For example, “feeling representation” is not mentioned until nearly the end of the paper, yet it is stated in the title. Additionally, although the title of the model (IMAC) emphasizes the modular architecture of the insula, it then de-emphasizes the other brain structures that play a role in interoceptive control and the visceral regulation.
- The authors could more clearly state which aspects of the model lack experimental evidence, while at the same time offering ways in which parts of this model could be directly tested using experimental tasks or manipulations. For example, how could researchers design studies to investigate the three specific levels of interoceptive processing in the subregions of the insula (interoceptive habit, model-based, vs. exploration, or mesa- vs. meso-conception)? Furthermore, how could dysfunction in these distinct processes be linked to what is known of different mental health disorders? This latter point will also help increase the clinical relevance of adopting such a

model and its distinctions, and how this model could help better understand disease processes, improve treatments, and predict patient outcomes.

Minor suggestions

- “Insula-visceral and somatic maps” section is vague, and a clearer description of topographic maps of the different sensations (thermal, somatosensory, visceral, etc.) could be provided.
 - o Recent work argues against a topographic representation for gustatory stimuli (see Avery 2021, “Against gustotopic representation in the human brain: there is no Cartesian restaurant”, *Curr Opin Physiol*, DOI: 10.1016/j.cophys.2021.01.005).
 - o Does “topographic representation of emotional, motivational, cognitive, and visceromotor processing” (page 6, line 11) mean that different emotions (i.e. sad, angry) or motivations (i.e. hunger, thirst) are linked to spatially adjacent but non-overlapping locations on the insula?
 - o Do “insula-visceral and insula-somatic maps” refer to a proposed generative map or model of the body that may be stored in the insula, or simply a topographic organization of the sensory information?
- The concept of allostasis should be mentioned earlier on in the paper, when discussing “adaptive interoceptive inference” and the active inference framework (Figure 2)
- Page 8 line 18: “neural” should be “neural structures”
- Page 11 line 20: “effect” should be “affect”

Optional suggestions

- The authors consider efferent, regulatory output under the definition of interoception, but could consider commenting on the distinction between afferent sensory and efferent regulatory arms.
- The introductory paragraph to begin the paper reads more like a list, and never defines interoception.
- Other cortical areas in addition to the insula can be outlined more fully, in relation to the processes they contribute to, rather than kept in one separate section towards the end. For example, other areas such as the cingulate cortex may contribute to the generation of visceromotor commands that can occur as part of motor, affective, and cognitive networks for the regulation of internal bodily states.
- In Figure 3, the authors could consider adding the dopaminergic areas that modulate the precision of signals.

===PREPARING YOUR MANUSCRIPT===

===PREPARING YOUR REVISION IN SCHOLARONE===

<https://royalsociety.org/journals/authors/author-guidelines/#data>. You should ensure that

you cite the dataset in your reference list. If you have deposited data etc in the Dryad repository, please include both the 'For publication' link and 'For review' link at this stage.

Author's Response to Decision Letter for (RSOS-201477.R0)

See Appendix A.

RSOS-220226.R0

Review form: Reviewer 2

Is the manuscript scientifically sound in its present form?

Yes

Are the interpretations and conclusions justified by the results?

Yes

Is the language acceptable?

Yes

Do you have any ethical concerns with this paper?

No

Have you any concerns about statistical analyses in this paper?

No

Recommendation?

Accept with minor revision (please list in comments)

Comments to the Author(s)

I think the manuscript has become much improved and the central argument much better delineated after the edits, and I appreciate the thoughtful and thorough responses by the authors to address all the reviewers' comments. I have a few remaining comments outlined below.

General comments:

- Although the “Evidence for the Model” section is a great addition to the paper, the authors could consider distributing it throughout the paper under the corresponding subsections rather than leaving it all to the end. For example, the anatomical modular structure of the model could then be followed by evidence for this anatomical arrangement, and so on.
- In the authors’ response to the reviewer, they explain that they have tried in the manuscript to conceptually separate the sensory arm (interoception) from the motor arm (visceral responses that are generated as a consequence of interoceptive predictions)... however, there are some inconsistencies, specifically when they state that interoception is analogous to motor control. Interoception may be the sensory arm to a predictive body regulation system that as a whole is analogous to motor control, but for the inner body as opposed to skeletal muscles.
- Although implied, the authors could explicitly state the intuitive (and theoretically necessary) link between actions – i.e. the somatomotor movements that constitute behaviours – and the visceromotor responses they require for energetic support. Somatomotor commands to skeletal muscles must be coupled with the appropriate visceromotor changes in body parameters to support blood flow, blood glucose, etc. Although these changes could happen retrospectively via reflexive autonomic loops, it is likely that the brain circuits responsible for the initiation of actions must also initiate the prospective interoceptive predictions that generate these visceral responses. This pairing of somatomotor and visceromotor outputs could be emphasized.
- While the central question of the paper is strong (why is a cortical representation of bodily signals necessary when the subcortical and brainstem structures can support survival?), the authors could better comment on whether conscious feelings are necessary for the higher-order interoceptive representations and policies. Theoretically, metaception and the associated prospective predictions could occur without subjective feelings of the body states, and much of this body regulation happens normally happens below the level of conscious awareness as it is (e.g. most people are not aware of their heartbeat under resting conditions). However, it could be that conscious feelings are a useful tool to promote the regulation of body states in addition to visceral responses (like putting on a sweater when cold before instigating shivering responses).
- I understand the concerns with length, but the authors could still offer some thoughts on what would count as sufficient evidence to show some of the components of their model. This could be achieved while maintaining brevity, using a couple sentences in each section or a summary table. For example, “if a study looked at this and found this, that would be compelling evidence to support this aspect of IMAC”. Furthermore, specifically what type of evidence would be needed to support the distinction between metaception and mesaception, and could studies be designed specifically to investigate one, the other, or where to draw the line between them?
- The authors could more directly address how this model compares with other theories and its importance within the literature, specifically the EPIC model proposed by Barret and Simmons (Nature Reviews Neuroscience 2015). This could be done by stating how the IMAC model is similar in terms of the emphasis on cytoarchitecture differences, but expands upon the EPIC model by including broader networks of brain regions. It could also draw upon some of the cytoarchitectonic detail of the EPIC model in terms of the cell types and information passing within the three main subregions.
- The authors seem to conflate the terms emotion and feeling; for example, page 1 line 15 they mention the “higher-order representation of conscious interoceptive feelings, which are built upon the basic emotions and underlying visceral processes”. They never offer an explicit definition of an emotion versus the feeling of an emotion or overarching affect, and this is important given the highly debated definition of an emotion. Additionally, while they maintain

that emotions are non-conscious, they state on page 15 line 1 that “feelings are under conscious and higher-order cognitive control”. While conscious control could affect a meta-awareness of the feeling, can it affect the generation of the feeling itself?

- The authors could provide a better definition of the following machine learning terms in the context of the IMAC model and how these terms apply to interoception: “model-based”, “interoceptive policies”, and “environmental state transitions”. What is the “goal of the agent” for interoceptive inference models? These are important aspects of the IMAC model and are mentioned numerous times, but if not familiar with machine learning literature I worry it may not be clear for some readers.

- o Additionally, the authors mention “interactions with, or sampling of the environment” in their definition of active inference on page 3, but they do not mention what “sampling of the environment” would look like for the inner body environment, as in interoceptive inference? In active inference, agents can choose behaviours that seek out more information about the environment, e.g. picking up an object to get a better visual... for the inner body, would this be akin to gating mechanisms that amplify or suppress interoceptive input to insular cortices?

- The authors comment numerous times on topographic representation of interoceptive information within the insula (e.g. page 7, “posterior to anterior” “topographic viscerosensory maps”), while also maintaining the integrated nature of such information (“information integration for awareness”, page 2; “integration and relay of interoceptive information” in the brainstem, page 5; hierarchical neurovisceral integration model, page 20). However, they do not comment on the potentially contrasting nature of these ideas, and what this could mean for the nature of the information reaching the insula... topographic representation implies the preservation of more or less parallel streams of sensory information ascending through the brainstem/thalamus relays to the insula, whereas integration implies the combining of information at lower levels or within the insula modular hierarchy. There are some opinions against simple spatial topographic mapping, instead opting for pattern-based representations; see Avery Current Opinion in Physiology 2001 for an example using taste.

- The figures have been improved for comprehensiveness and completeness, and Figures 1A and 1B do give the reader an indication of just how complex the interoceptive system is with all the different body systems, pathways, and brain regions/connections that are involved. However, Figure 1B still has too much detail that makes it visually confusing. With so many crossing arrows, it is hard to follow the connections and it’s not clear which are wanting to be highlighted. The authors could consider better ways to combine, consolidate, and simplify some of the arrows, and if necessary split up the figure further into one overall figure with then sub-figures providing more detail on the connectivity of different parts (but not all the detail of all the parts on just one figure).

Minor comments:

- The authors do not differentiate between the ANS and the interoceptive nervous system (INS). Additionally, although they make comparisons to exteroception in terms of active inference processes, they do not comment on the differences in the nature of interoceptive information compare to exteroception – see Carvalho and Damasio, BioEssays 2021 (specifically Table 1).

- The authors associate allostatic control with metaception and the cortical regions supporting this. To me, this implies that there is no evidence of allostatic control within the brainstem and other subcortical regions, that these would be strictly retrospective homeostatic control... is this the case?

- The first couple sentences of the “IMAC Implications for Understanding Depression” seem slightly off-topic.. it may be better to begin by stating how interoceptive dysregulation is implicated in the pathophysiology of depression, before mentioning the public health importance of depression and its prominence given the COVID-19 pandemic.
- The authors could consider citing Hassanpour et al. *Neuropsychopharmacology* 2018 or Teed et al. *Jama Psychiatry* 2022 when referencing tasks that deliver unpredictable interoceptive stimulation (page 21 line 23). These papers increased cardiorespiratory arousal during scanning via intravenous infusions of the adrenaline analogue isoproterenol, and found regions of the insula to be primarily responsive to this peripheral stimulation.
- Grammatical errors in the following sentences:
 - o Use of semicolons: page 2 line 9, page 8 line 14, page 14 line 19.
 - o Page 9 line 4: “uses” should be “use”
 - o Page 19 line 29: should “sufficient” be “insufficient”?
 - o Page 20 line 29: should be “number of synaptic connections”

*note: page numbers are taken from upper right corner of manuscript, not the whole pdf document including reviewer comments and responses. Line numbers are taken from the inner line numbers closest to the lines in the pdf.

Decision letter (RSOS-220226.R0)

Dear Dr Fermin

On behalf of the Editors, we are pleased to inform you that your Manuscript RSOS-220226 "An Insula Hierarchical Network Architecture for Active Interoceptive Inference" has been accepted for publication in Royal Society Open Science subject to minor revision in accordance with the referees' reports. Please find the referees' comments along with any feedback from the Editors below my signature.

Please submit your revised manuscript and required files (see below) no later than 7 days from today's (ie 10-May-2022) date. Note: the ScholarOne system will 'lock' if submission of the revision is attempted 7 or more days after the deadline. If you do not think you will be able to meet this deadline please contact the editorial office immediately.

on behalf of Dr Anastasia Christakou (Associate Editor) and Essi Viding (Subject Editor)
 openscience@royalsociety.org

Associate Editor Comments to Author (Dr Anastasia Christakou):
 Associate Editor

Comments to the Author:

I was very pleased to see the revised manuscript, which has made great use of the thorough and thoughtful reviewers' comments. Thank you for your careful attention and the considerable effort you have made in improving the work.

There are a few remaining issues raised by your reviewer that we feel should be addressed to help improve the manuscript further.

Alongside the well detailed suggestions of the reviewer, I strongly encourage you to pay particular attention to interventions that will make the paper accessible to a wider readership, such as clarifying (your use of) specialist or ambiguous terms and editing the schematic depiction of the circuitry in figure 1B (e.g.: you may wish to sacrifice detail for clarity of communication on the basis of the message that the figure is designed to support). I would also encourage you to favour clarity and completeness over brevity when considering your treatment of the comments about clarifying the model's relationship with existing models, and providing examples of testable predictions/falsifications.

Congratulations on what will be an important paper. I look forward to receiving your final treatment.

Reviewer comments to Author:

Reviewer: 2

Comments to the Author(s)

I think the manuscript has become much improved and the central argument much better delineated after the edits, and I appreciate the thoughtful and thorough responses by the authors to address all the reviewers' comments. I have a few remaining comments outlined below.

General comments:

- Although the "Evidence for the Model" section is a great addition to the paper, the authors could consider distributing it throughout the paper under the corresponding subsections rather than leaving it all to the end. For example, the anatomical modular structure of the model could then be followed by evidence for this anatomical arrangement, and so on.

- In the authors' response to the reviewer, they explain that they have tried in the manuscript to conceptually separate the sensory arm (interoception) from the motor arm (visceral responses that are generated as a consequence of interoceptive predictions)... however, there are some inconsistencies, specifically when they state that interoception is analogous to motor control. Interoception may be the sensory arm to a predictive body regulation system that as a whole is analogous to motor control, but for the inner body as opposed to skeletal muscles.

- Although implied, the authors could explicitly state the intuitive (and theoretically necessary) link between actions – i.e. the somatomotor movements that constitute behaviours – and the visceromotor responses they require for energetic support. Somatomotor commands to skeletal muscles must be coupled with the appropriate visceromotor changes in body parameters to support blood flow, blood glucose, etc. Although these changes could happen retrospectively via reflexive autonomic loops, it is likely that the brain circuits responsible for the initiation of actions must also initiate the prospective interoceptive predictions that generate these visceral responses. This pairing of somatomotor and visceromotor outputs could be emphasized.

- While the central question of the paper is strong (why is a cortical representation of bodily signals necessary when the subcortical and brainstem structures can support survival?), the authors could better comment on whether conscious feelings are necessary for the higher-order interoceptive representations and policies. Theoretically, metaception and the associated prospective predictions could occur without subjective feelings of the body states, and much of this body regulation happens normally happens below the level of conscious awareness as it is (e.g. most people are not aware of their heartbeat under resting conditions). However, it could be that conscious feelings are a useful tool to promote the regulation of body states in addition to visceral responses (like putting on a sweater when cold before instigating shivering responses).

- I understand the concerns with length, but the authors could still offer some thoughts on what would count as sufficient evidence to show some of the components of their model. This could be achieved while maintaining brevity, using a couple sentences in each section or a summary table. For example, “if a study looked at this and found this, that would be compelling evidence to support this aspect of IMAC”. Furthermore, specifically what type of evidence would be needed to support the distinction between metaception and mesaception, and could studies be designed specifically to investigate one, the other, or where to draw the line between them?

- The authors could more directly address how this model compares with other theories and its importance within the literature, specifically the EPIC model proposed by Barret and Simmons (Nature Reviews Neuroscience 2015). This could be done by stating how the IMAC model is similar in terms of the emphasis on cytoarchitecture differences, but expands upon the EPIC model by including broader networks of brain regions. It could also draw upon some of the cytoarchitectonic detail of the EPIC model in terms of the cell types and information passing within the three main subregions.

- The authors seem to conflate the terms emotion and feeling; for example, page 1 line 15 they mention the “higher-order representation of conscious interoceptive feelings, which are built upon the basic emotions and underlying visceral processes”. They never offer an explicit definition of an emotion versus the feeling of an emotion or overarching affect, and this is important given the highly debated definition of an emotion. Additionally, while they maintain that emotions are non-conscious, they state on page 15 line 1 that “feelings are under conscious and higher-order cognitive control”. While conscious control could affect a meta-awareness of the feeling, can it affect the generation of the feeling itself?

- The authors could provide a better definition of the following machine learning terms in the context of the IMAC model and how these terms apply to interoception: “model-based”, “interoceptive policies”, and “environmental state transitions”. What is the “goal of the agent” for interoceptive inference models? These are important aspects of the IMAC model and are mentioned numerous times, but if not familiar with machine learning literature I worry it may not be clear for some readers.

o Additionally, the authors mention “interactions with, or sampling of the environment” in their definition of active inference on page 3, but they do not mention what “sampling of the environment” would look like for the inner body environment, as in interoceptive inference? In

active inference, agents can choose behaviours that seek out more information about the environment, e.g. picking up an object to get a better visual... for the inner body, would this be akin to gating mechanisms that amplify or suppress interoceptive input to insular cortices?

- The authors comment numerous times on topographic representation of interoceptive information within the insula (e.g. page 7, “posterior to anterior” “topographic viscerosensory maps”), while also maintaining the integrated nature of such information (“information integration for awareness”, page 2; “integration and relay of interoceptive information” in the brainstem, page 5; hierarchical neurovisceral integration model, page 20). However, they do not comment on the potentially contrasting nature of these ideas, and what this could mean for the nature of the information reaching the insula... topographic representation implies the preservation of more or less parallel streams of sensory information ascending through the brainstem/thalamus relays to the insula, whereas integration implies the combining of information at lower levels or within the insula modular hierarchy. There are some opinions against simple spatial topographic mapping, instead opting for pattern-based representations; see Avery Current Opinion in Physiology 2001 for an example using taste.

- The figures have been improved for comprehensiveness and completeness, and Figures 1A and 1B do give the reader an indication of just how complex the interoceptive system is with all the different body systems, pathways, and brain regions/connections that are involved. However, Figure 1B still has too much detail that makes it visually confusing. With so many crossing arrows, it is hard to follow the connections and it's not clear which are wanting to be highlighted. The authors could consider better ways to combine, consolidate, and simplify some of the arrows, and if necessary split up the figure further into one overall figure with then sub-figures providing more detail on the connectivity of different parts (but not all the detail of all the parts on just one figure).

Minor comments:

- The authors do not differentiate between the ANS and the interoceptive nervous system (INS). Additionally, although they make comparisons to exteroception in terms of active inference processes, they do not comment on the differences in the nature of interoceptive information compare to exteroception – see Carvalho and Damasio, BioEssays 2021 (specifically Table 1).

- The authors associate allostatic control with metaception and the cortical regions supporting this. To me, this implies that there is no evidence of allostatic control within the brainstem and other subcortical regions, that these would be strictly retrospective homeostatic control... is this the case?

- The first couple sentences of the “IMAC Implications for Understanding Depression” seem slightly off-topic.. it may be better to begin by stating how interoceptive dysregulation is implicated in the pathophysiology of depression, before mentioning the public health importance of depression and its prominence given the COVID-19 pandemic.

- The authors could consider citing Hassanpour et al. Neuropsychopharmacology 2018 or Teed et al. Jama Psychiatry 2022 when referencing tasks that deliver unpredictable interoceptive stimulation (page 21 line 23). These papers increased cardiorespiratory arousal during scanning via intravenous infusions of the adrenaline analogue isoproterenol, and found regions of the insula to be primarily responsive to this peripheral stimulation.

- Grammatical errors in the following sentences:

o Use of semicolons: page 2 line 9, page 8 line 14, page 14 line 19.

o Page 9 line 4: “uses” should be “use”

- o Page 19 line 29: should “sufficient” be “insufficient”?
- o Page 20 line 29: should be “number of synaptic connections”

*note: page numbers are taken from upper right corner of manuscript, not the whole pdf document including reviewer comments and responses. Line numbers are taken from the inner line numbers closest to the lines in the pdf.

===PREPARING YOUR MANUSCRIPT===

one version should clearly identify all the changes that have been made (for instance, in coloured highlight, in bold text, or tracked changes);
 a 'clean' version of the new manuscript that incorporates the changes made, but does not highlight them. This version will be used for typesetting.

===PREPARING YOUR REVISION IN SCHOLARONE===

Please ensure that you include a summary of your paper at the 'Type, Title, & Abstract' step. This should be no more than 100 words to explain to a non-scientific audience the key findings of your

research. This will be included in a weekly highlights email circulated by the Royal Society press office to national UK, international, and scientific news outlets to promote your work. An effective summary can substantially increase the readership of your paper.

-- If you are requesting an article processing charge waiver, you must select the relevant waiver option (if requesting a discretionary waiver, the form should have been uploaded, see 'File upload' above).

-- If you have uploaded any electronic supplementary (ESM) files, please ensure you follow the guidance at <https://royalsociety.org/journals/authors/author-guidelines/#supplementary-material> to include a suitable title and informative caption. An example of appropriate titling and captioning may be found at https://figshare.com/articles/Table_S2_from_Is_there_a_trade-off_between_peak_performance_and_performance_breadth_across_temperatures_for_aerobic_scope_in_teleost_fishes_/3843624.

Author's Response to Decision Letter for (RSOS-220226.R0)

See Appendix B.

Decision letter (RSOS-220226.R1)

Dear Dr Fermin:

I am pleased to inform you that your manuscript entitled "An Insula Hierarchical Network Architecture for Active Interoceptive Inference" is now accepted for publication in Royal Society Open Science.

Please remember to make any data sets or code libraries 'live' prior to publication, and update any links as needed when you receive a proof to check - for instance, from a private 'for review' URL to a publicly accessible 'for publication' URL. It is also good practice to add data sets, code and other digital materials to your reference list.

Royal Society Open Science is a fully open access journal. A payment may be due before your article is published. Our partner Copyright Clearance Centre will contact the corresponding author about your open access options (if you have any queries regarding fees, please see <https://royalsocietypublishing.org/rsos/charges> or contact authorfees@royalsociety.org).

on behalf of Dr Anastasia Christakou (Associate Editor) and Dr Essi Viding (Subject Editor).

Follow Royal Society Publishing on Twitter: @RSocPublishing
Follow Royal Society Publishing on Facebook:
<https://www.facebook.com/RoyalSocietyPublishing/>
Read Royal Society Publishing's blog:
<https://royalsociety.org/blog/blogsearchpage/?category=Publishing>

Appendix A

We are extremely grateful to the reviewers' comments and suggestions to improve the quality of our manuscript.

Several steps were taken during the revision of the manuscript, including reorganization of the presentation of the contents, the use of terminology, technical issues related to brain connectivity and improvement of the figures.

We tried our best to implement all reviewers' suggestions.

It's our belief that the current version reflects their suggestions with clearer ideas and follow of the hypothesis developed in the manuscript.

Comments to the Author(s)

REVIEWER 1

General comments.

The notion that insula is organized into hierarchical modules needs to be clarified

!) what is evidence that these are modules i.e. serving as discrete functional units

2) what is the evidence that they are hierarchical

To address this requires a deep analysis of the functional architecture of insular cortex. Several observations support modularity e.g. topographical segregation of taste cortex or inputs from heart versus gut suggest functional modularity but such observations may also work against notion of hierarchical organization I,e, not all representations start in project granular regions and project sequentially to agranular insula.

Sequential projection within insula for posterior to anterior is central to Craig's model of insular interoceptive representation and ultimately conscious feelings, and this was informed by his own anatomical studies. Similarly, neuroanatomical work of Helen Barbas has informed Barratt and colleagues own model of insula in interoception predictive representation.

Henry Evrard has undertaken the most comprehensive cytoarchitectonic studies of primate insula, identifying at least 22 subregions. And mapping inputs and outputs to each region. This should provide the anatomical context for understanding not only how insular organization fits the notion of an interoceptive hierarchy proposed in this paper.

1. AUTHOR'S RESPONSE TO THE ABOVE GENERAL COMMENTS:

In order to address the issue of insula modularity in more detail, we added several lines supporting our view of a modular functional and anatomical organization of the insular cortex:

Local anatomical organization (page 6, line 8-13)

TEXT: The granular insula (gINS) is one of the three insular modular structures identified based on the expression of a cellular layer IV, containing predominantly granular cells: the agranular insula (aINS), located in a ventral anterior position and lacking a granular layer IV, the gINS, located in the most dorsal posterior position, with a fully developed layer IV, and the dysgranular insula (dINS), located between the aINS and gINS, but with an underdeveloped layer IV [19,43–48].

Pathways linking the insula to the striatum and dopaminergic system (page 7, lines 8-14)

TEXT: The aINS makes connections primarily with the ventral striatum (vStriatum), the dINS with the dorsomedial striatum (mStriatum), and the gINS with the dorsolateral posterior striatum (pStriatum) [19]. Striatum sub-regions also receive topographical input from the dopaminergic system, with the vStriatum receiving its main dopaminergic input from the ventral tegmental area (VTA), and the mStriatum and pStriatum from the medial and ventro-lateral substantia nigra complex, respectively [74].

Cerebral cortex (page 6, line 32, to page 7, lines 1 to 8)

TEXT: Cortical regions involved in primary sensory (somatosensory and auditory cortices), motor (primary and supplementary motor areas) and environmental information processing (superior and inferior parietal cortices) project predominantly to the gINS [13,66]. It is important to note here that the inferior parietal cortex contains the supramarginal and angular gyri that form the temporo-parietal junction (TPJ), a cortical complex implicated in social cognition [67–72]. The dINS, on the other hand, receives anatomical input from the dorsolateral prefrontal cortex (Brodmann areas 45 and 46) [43–45,73]. In contrast, the aINS has predominant connections with the ACC, ventral anterior PFC, ventromedial orbitofrontal cortex, amygdala and hippocampal complex [13,43–45].

Thalamus (page 6, lines 30-32)

TEXT: It has been shown in monkeys that the aINS, dINS and gINS make bidirectional connections with the thalamic nuclei, from which they receive visceral information [65].

Brainstem (page 7, lines 14 to16)

TEXT: Neuroanatomical evidence also suggests that each insula module has direct projections to brainstem visceral motor nuclei that modulate visceral and physiological processes, such as gastric functions, heart rate, blood pressure, pain and hormone secretion [75–84].

We included one study showing that the modular organization of the insula into anterior and posterior regions is present at birth (page 6, line 13-14).

TEXT: The modular organization of the human insular cortex (posterior and anterior segments) is already present in neonates [49].

Topographic representation of interoceptive information (page 7, lines 19-24)

TEXT: The insula contains topographic viscerosensory maps located predominantly in its granular sub-region, extending in a posterior to anterior direction, that represent vestibular, nociceptive, thermoreceptive, visceral, and gustatory information [13,42]. Stimulation studies in humans have revealed a similar topographic representation of pain, thermal, somatosensory, visceral, vestibular, and gustatory information [85–88], although somatosensory information seems to be represented throughout the insular cortex [89,90].

Functional modularity (page 7, lines 24-28)

TEXT: Human neuroimaging studies also show a functional modular organization that maps onto the anatomical modular organization of the insula, with visceromotor information represented in the gINS, emotional and motivational information represented in the ventral aINS, and cognitive information represented in the dorsal anterior insula, including the dINS [91–96].

We recognize Evrard's elegant works on the fine parcellation of the insular architecture. In fact, as described in Nieuwenhuys (2012), several previous works have also provided finer parcellations of the insula. Since the exact functions of such parcellations have yet to be elucidated, we decided to use (page 6, lines 14-18) to limit our opinion on the more well-known three-insula modular organization (granular, dysgranular, agranular).

TEXT: Other studies have suggested a more detailed parcellation and modular organization of the insula, sometimes sub-dividing the insular cortex into 31 sub-regions [13,42,47,48,50–53]. However, since the precise roles of these finer insular parcellations have yet to be determined, the IMAC model considered here will focus on the general three-insula modular cytoarchitecture (gINS, dINS and aINS).

We understand that the insular cortex receives interoceptive information from multiple parallel pathways. To address this issue, we updated Figure 1 to include several ascending pathways transmitting

information to the insula. Our opinion, however, on the hierarchical organization of the insula modules emphasizes the ascending interoceptive pathways through the thalamic visceral nuclei, which innervates the granular insula and from there propagates interoceptive information to the dysgranular and agranular regions of the insula. Although the view between modular organization and hierarchical structure might seem incongruent at first, the hierarchical organization of modular structures has been recognized by several works on the basal ganglia, visual, temporal and prefrontal cortices.

TEXT: Local anatomical connectivity among the three insula sub-regions is very distinct, with reciprocal connections between the gINS and dINS and between the dINS and aINS, and modest aINS fiber output to the gINS [43–45]. This pattern of connectivity suggests a hierarchical organization for interoceptive information processing within the insular cortex that is similar to the hierarchical organization observed in other sub-cortical and cortical systems that subservise perceptual, action control, and higher-order cognitive processes, including the basal ganglia, and the visual, temporal and prefrontal cortices [54–64].

The introduction jumps straight into a problem area: The distinction between efferent and afferent innervation of internal organs. Simply, the autonomic nervous system (ANS) is typically viewed as peripheral and efferent (sympathetic and parasympathetic nerves). This is distinct from interoception where viscerosensory afferent nerves carry interoceptive information centrally from visceral interoceptors. Interoceptive pathways may share nerves with autonomic efferents but not always. They also clearly do inform “autonomic” control (as do other channels of sensory information). However, if the authors want to conflate afferent interoception with efferent autonomic control they need to make a stronger case for doing so. Line 6-7 effectively says ‘the motor system is responsible for transmitting somatosensory information from brain to body... this may be true at a stretch, in active inference terms, but the distinct afferent pathway mediating this is missing. Extending this to say the ANS is responsible for thermoregulatory hormonal and immune responses is a further stretch give humoral sampling by paraventricular organs is a different type of channel from central neural signalling from the periphery.

TL:DR I take issue with the broad redefinition of the autonomic nervous system and its conflation with interoceptive signalling.

We strongly agree with the reviewer’s comments on our stretching of the concept of ANS and their sensory and motor control pathways. In order to fully address this issue, we reorganized the structure of the manuscript to better fit the reviewer’s suggestions.

In order for the audience to understand that visceral information arrives from the visceral organs through spinal pathways and non-spinal cranial nerve pathways, we kept the general organization of the sympathetic and parasympathetic pathways. Our focus, however, changed to the ascending interoceptive pathways to the brainstem and insular cortex. Keeping a discussion on the motor pathway would add nothing to our discussion. Also, we believe to have removed any direct link between ANS and control of thermoregulation, hormonal and immune functions.

Figure 1 has also been improved to reflect in more detail several ascending interoceptive pathways linking brainstem nuclei to subcortical regions, hypothalamus, thalamus and insula.

Paragraphs 1 and 2 (page 4, lines 28-32; page 5, lines 1-32) of the second section of the manuscript (Insula neural architecture) presents the changes describe above.

The second paragraph of the introduction sets up a straw man: the premise that insula is the main target for interoceptive afferents: it is certainly a target for projection from a thalamic area that receives interoceptive information, but one might argue brainstem nuclei notably the NTS, are the main target of interoceptive afferents. That aside, in this opinion piece, perhaps there needs to be an early, stronger definition of interoception and what the authors think it is for (i.e. what is the question the new model is trying to address). Statements like 'interoceptive adaptive behavior' can then be better understood.

The second paragraph (page 2, lines 14-25) was re-organized to reflect the reviewer's comment. In this paragraph, we set the context to pose the problem for which our model may provide an answer. First, we provide a general view that the brainstem and its nuclei receive ascending interoceptive information. Next, we mention that the anterior cingulate cortex (ACC) and insula are the main targets of thalamic interoceptive information. However, we mention that the insula local organization and connectivity patterns may play a unique role in interoceptive information processing which has yet to be explained. The problem we pose is that if the brainstem nuclei are capable to generate all basic visceral and physiological adjustments to maintain survival functions, why does the cerebral cortex need a cortical insular representation of interoceptive information? (page 2, lines 31-32, pages 3, lines 1-5)

UPDATED TEXT

In the brain, the brainstem is the region that receives direct interoceptive information ascending from the visceral systems [1]. The brainstem contains several nuclei that receive direct ascending interoceptive information, such as nucleus tractus solitarius (NTS), the medullary reticular formation, the parabrachial nucleus (PBN), the periaqueductal gray area (PAG), and that are thought to generate innate, hard-wired, visceral and hormonal responses to maintain all physiological survival functions and to cope with the demands imposed by the body and the environment [1,12]. In the cerebral cortex, the insula and anterior cingulate cortex (ACC) are the main targets of interoceptive afferents arriving from the visceral systems through thalamic nuclei [3,5,13–17]. However, while the ACC is a predominantly agranular structure, i.e., it lacks a granular layer IV, the insula has a more complex organization with a topographic representation of visceral processes and three sub-regions with distinct levels of laminar granularity, distribution of acetylcholinergic receptors, and patterns of local, cortical, subcortical and brainstem connectivity [13,17–19] (Figure 1). These cytoarchitectonic and anatomical connectivity features suggest a central position and specialization of the insular cortex in processing interoceptive information. This convergence of cortical and interoceptive information upon the insular cortex has led to influential hypotheses concerning its potential roles in interoceptive prediction [14], information integration for awareness [20,21], emotional awareness [22], and interoceptive inference and emotion [23,24]. Despite the elegance and appeal of these hypotheses, no one has yet explained why the brain needs a cortical insular representation of visceral processes, given that the multiple brainstem nuclei are sufficient to generate all necessary visceral and physiological adjustments to maintain the body's survival functions. Furthermore, it remains unclear how insular functions interact with—and are modulated by—input from cortical and subcortical systems and from neuromodulatory systems, and whether such interactions underwrite survival.

Figure 1 b is puzzling: Leaving aside my view that it doesn't make functional sense to

speak of parasympathetic and sympathetic afferents (I presume the authors are referring to viscerosensory afferents to brain respectively via vagus /glossopharyngeal nerve and spinal cord – the later encompassing presumably sacral parasympathetic afferents), the authors appear to be proposing that there is a direct (non-synapsing) neural connection from some viscera to granular insula and that all NTS information to thalamus relays via parabrachial nucleus hypothalamus where and PAG (really). Obligatory pontine relays from NTS are a feature of rodents but not primates.

The reviewer is right, our goal was to describe viscerosensory afferents. Thus, we updated Figure 1 in several ways to reflect that. Figure 1A still maintains the separation between sympathetic and parasympathetic systems for the audience to understand that viscerosensory information reaches the brain via spinal cord and cranial nerve pathways. At the brain level, although not in the detail, the figure shows the NTS as the main nuclei receive ascending interoceptive information from the visceral organs. This information is either transmitted directly to the thalamus from the NTS or via brainstem nuclei, and from brainstem nuclei to the insula. We added Figure 1B and 1C. In Figure 1B, we focused on the ascending interoceptive pathways to the brainstem from the body and head. We also highlight the direct projections from the NTS to neuromodulatory systems (serotonergic, dopaminergic, noradrenergic and cholinergic) and how their direct connections to the cerebral cortex and insula. The dotted colored lines reflect possible direct projections from the NTS to the insular cortex, but this pathway wasn't stressed in the manuscript as further research is needed to validate these anatomical findings. Other brainstem nuclei that also receive visceral input were omitted. Figure 1C addressed some possible predominant cortical connections of each insula module.

The description of sympathetic and parasympathetic (efferent) pathways in the periphery is perhaps not central to the proposed model, but the statement that postganglionic sympathetic innervation of smooth muscle is cholinergic (P4 line 17,18) is wrong. Noradrenaline is the postganglionic neurotransmitter for all end organs with some few exceptions, notably sudomotor glands in adult humans. Circulating adrenaline thus also acts on the majority of sympathetically innervated organs. The parasympathetic innervation of the pelvis is not addressed for completion. The statement “most visceral information , reaches the brain via the vagus nerve” P4 line 31/32) needs some kind of quantitative evidence i.e. a primary source, as I would dispute this based on observation from vagotomy (which, as an aside, has also been used in management of visceral pain P5 l2)

We removed any unnecessary information on the sympathetic and parasympathetic systems which would not contribute to the main ideas developed in this opinion.

P5 l8-10 this description of central pathways to thalamus “before arriving at the insular cortex” is superficial; Craig has provided more detail about this e.g. how representations are mapped into different regions of thalamus, and he also identifies other thalamocortical projections (e.g. to cingulate of this information).

We provided an answer to this point above. In the manuscript, we focused on cortical regions receiving interoceptive information via the thalamus. Here, we recognized both the ACC and insula. However, given that the local cytoarchitecture of the ACC is quite simple (agranular), the opinion developed here focused on the insular cortex given its more widely recognized cytoarchitecture (agranular, dysgranular, granular) and well-studied topographic representation of interoceptive information (e.g. temperature, visceral, pain).

P5 I13-24 insula hierarchical organization: This section is quite critical to the model and should draw from Everard's work rather than the older papers cited. I am doubtful if virus tracing is the basis to definition of 3 larger cytoarchitectonic regions within insula, but may inform connections between regions. As noted above there is not up to date information in primate brain about this. The arguments about cortical connectivity as a hierarchy, typically refers to input (e.g. granule cell) and output layers of cortex – what is the evidence here for hierarchy. "Reciprocal connections between dINS and gINS suggest no hierarchy between these two regions, is this correct?"

We understand that the insular cortex receives interoceptive information from multiple parallel pathways. To address this issue, we updated Figure 1B to include several ascending pathways transmitting interoceptive information to the insula. Our opinion, however, on the hierarchical organization of the insula modules emphasizes the ascending interoceptive pathways through the thalamic visceral nuclei, which innervates the granular insula and from there propagates interoceptive information to the dysgranular and agranular regions of the insula. This view is recognized in Craig's works as well as in Everard's experimental work and review papers.

The view that modular organization and hierarchical structure might seem incongruent. Here, we adopt the view that has also been adopted by other works recognizing the hierarchical organization of modular structures found in the structure of the basal ganglia, visual, temporal and prefrontal cortices. We recognize that interoceptive information from the thalamus propagates in a posterior granular region to a ventral anterior agranular region of the insular cortex. As interoceptive information travels through the insula, it is further processed locally and integrated with information arriving from other cortical and subcortical regions. This integration increases the complexity of the information encoded in the insular cortex, from a visceral- and sensory-based representation located in the granular insula, to cognitive, motivation and emotion driven representation located in the dysgranular and agranular insula. Thus, our view is no different from that proposed by Craig. Furthermore, forward and backward connections have been recognized as hallmark of the hierarchical organization observed in the structure of the prefrontal, visual and temporal cortices. We believe that the forward and backward connections among the insula modules is similar to that observed in these other cortical systems and to the predictive coding hypothesis on which our model is based.

The text below, incorporated to the manuscript, tries to succinctly convey this idea (page 4, lines 21-27).

TEXT: Local anatomical connectivity among the three insula sub-regions is very distinct, with reciprocal connections between the gINS and dINS and between the dINS and aINS, and modest aINS fiber output to the gINS [43–45]. This pattern of connectivity suggests a hierarchical organization for interoceptive information processing within the insular cortex that is similar to the hierarchical organization observed in other sub-cortical and cortical systems that subserves perceptual, action control, and higher-order cognitive processes, including the basal ganglia, and the visual, temporal and prefrontal cortices [54–64].

P5 The insular dopaminergic connection needs to be better described i.e. what cells do they innervate, and what is their function? In rodents agranular insula has a dopamine input (like OFC), with some identified functions. Dopamine may be especially relevant to the proposed IMAC model, but other neuromodulators (e.g. noradrenaline from locus ceruleus) have been implicated in similar models. A case needs to be made for

selecting dopamine among for comment.

Thanks to the reviewer's comments, we made several improvements to suggest possible of dopamine and acetylcholine on the processing of interoceptive information in collaboration with the insula. We also updated Figure 1B and Figure 3C to include connections between the NTS and other neuromodulatory systems (serotonin and noradrenaline), but did not elaborate on this interoception-neuromodulatory pathway given the length of the manuscript. Our future work (in preparation) will expand our view on this topic. The texts below have been incorporated to the manuscript to suggest candidate roles of dopamine (one of our simulation studies has been added) and acetylcholine on their relationship with the insula.

TEXT (page 6, lines 1-6): Finally, NTS interoceptive signals can quickly exert influences over not only the insular cortex but the whole cerebral cortex via its direct projections to neuromodulatory systems that bypass other brainstem nuclei, including the dopaminergic system (ventral tegmental area and substantia pars compacta), the noradrenergic system (locus coeruleus), the serotonergic system (dorsal raphe and medial raphe), and cholinergic system (nucleus basalis of Meynert) (Figure 1B) [39].

In the text below, we extend the well-known functions of the PFC-striatum-dopamine loops in action selection and confidence (dopamine role) of hypothetical or selected actions to the insula-striatum-dopamine loops.

TEXT (page 10, lines 14-18): Under this view, while the PFC-striatum-dopamine loops support learning and optimization of action selection and other decision-making processes, insula-striatum-dopamine loops are concurrently actively seeking to generate optimal interoceptive predictions to generate visceral responses necessary to achieve the physiological demands of desired actions, behaviors and mental processes.

TEXT (page 10, 27-28): A gINS interoception policy is selected by dopaminergic signals arriving from the ventro-lateral SN complex onto the pStriatum that evaluate the degree of confidence in the appropriate interoceptive predictions, generated by the gINS. This fits comfortably with the role of dopamine as encoding the precision or salience of action-pointing representations [142].

TEXT: In the context of active inference, the striatum is responsible for selection of cortical representations based on precision signals mediated by dopaminergic neurons [28]. Based on the above findings, the IMAC model hypothesizes that dopaminergic input to the striatum serves to signal the precision of interoceptive predictions arriving from the insular cortex—by selectively increasing the postsynaptic sensitivity to insular afferents: please see [178] for a simulation of this dopaminergic selection, in the context of predictive coding and hierarchical motor control in Parkinson's disease. The exact role of striatum direct and indirect pathways on interoceptive processes has yet to be determined. One possibility is that striatum direct and indirect pathways are concurrently engaged in learning and selection of interoceptive representations that promote survival and suppression of interoceptive representations that may cause harm, respectively

A role for acetylcholine was proposed as in the text below:

TEXT (page 13, lines 9-20): Acetylcholine increases synaptic transmission in the thalamus, hippocampus, and PFC, making the activity of neurons in these regions more responsive to synaptic input from other brain areas, and facilitating experience-dependent learning [164–166]. The IMAC model hypothesizes that the higher density of acetylcholine receptors in the aINS and dINS endows these regions with greater capability to flexibly modify or learn metaceptive contingencies for interoceptive prediction and to create novel interoceptive mappings. In contrast, activity of the gINS can exert direct influence on visceral interoceptive functions, such as generation of cardiac arrhythmias [86,167]. The IMAC model

hypothesizes that the lower density of acetylcholine receptors in the posterior gINS reflects the stability of interoceptive predictions that directly impact visceral survival functions. In short, the neuroplastic potential in the interoceptive (insular) hierarchy increases with hierarchical depth, enabling the learning of deep generative models of the embodied self.

IMAC Model P8 – the first paragraph could be more explicit in associating active inference with) action or internal efferent responses to differentiate active inference within general predictive coding models. This may mean a clearer definition of policies (line 2).

In order to fulfill this suggestion, we improved the third paragraph of page 3 of the Introduction as in the text below.

TEXT (page 3, lines 17-32): According to the theory of active inference, the brain uses internal generative models, acquired through experience, to continuously generate descending or top-down predictions of expected sensory data [25–27]. In active inference, the goal of the agent is to find optimal policies that minimize free-energy, or prediction errors, between predicted and actual sensory input generated by the agent's interactions with, or sampling of, the environment [28,29]. The theory of allostasis proposes a similar predictive process for regulation of physiological states of the body [7,30–32]. The allostasis model suggests that the brain uses innate or learned prior knowledge of physiological states, e.g., glucose levels and heart rate, to predict future physiological states and thereby pre-emptively preclude deviations or prediction errors from homeostatic setpoints. Based on the allostatic reading of active inference [23,33,34], the IMAC model hypothesizes that the hierarchical modular cytoarchitecture of the insular cortex—supported by its parallel neural networks with the PFC and striatum-dopaminergic and acetylcholinergic systems – is specialized in higher-order interoceptive inference, hereby called metaception. That is, it specializes in the construction of cortical representations of lower-order, innate, interoceptive representations, herein called mesaception, located in subcortical, i.e., amygdala and hypothalamus, and brainstem nuclei. In the IMAC framework, metaception is regarded as an evolved cortical capacity to generate flexible higher order interoceptive predictions that are concurrently computed with action predictions that seek to maximize an individual's long-term fitness in her interactions with the environment. In other words, metaception is the perceptual synthesis (c.f., Bayesian belief updating) that furnishes high-level representations (c.f., feelings), which can predict lower level interoceptive representations and accompanying responses in the motor and autonomic domain.

This model of insular information flow is similar to the proposed bottom-up model of Craig's. And, is the cortical connectivity in insula specialized, i.e. different from elsewhere in the brain, for predictive coding? Is there empirical evidence for interoceptive prediction error computations within dINS?

Yes, we agree with the reviewer that the bottom-up feature of the IMAC model is similar to that proposed by Craig. However, the IMAC model also incorporates a top-down feature in which modules of higher hierarchy seek to train the modules of lower hierarchy. Thus, when the agranular and dysgranular insula modules work to learn novel interoceptive representations, their learning signals teach the granular insula to store interoceptive policies which can be quickly relied upon when the same behavior context is encountered. This view is an extension of the neural findings of studies on motor learning, sequential action learning, and decision making which revealed that the motor cortex and SMA store motor memories, the DLPFC implements model-based control and the VMPFC implements motivational and exploratory control.

Future works need to provide strong evidence that the insular cortex implements predictive processes and Bayesian interoceptive belief update. However, in recent years, a few studies in humans and animals

have started to support this view. We added this short sentence in the manuscript to convey this limitation:

TEXT (page 19, lines 12-14): Evidence that the insula generates interoceptive predictions and uses Bayesian belief updating is still scarce but has started to emerge from human and animal experiments [247–252].

Figure 2 links articulates the IAC model with the proposal of parallel hierarchies for “interoceptive habit, model-based and exploration”. Again there is assumption that the selected interregional connectivity supports hierarchical predictive processing: It is noteworthy how many brain regions are included in the model.

The IMAC model as described in the text is insightful. The point regarding similarity to known prefrontal striatal “loops” are well made, and the narrative how these parallel systems might work together is compelling and overcomes many limitations of earlier models that tend to over-generalize. The notion of model-based strategy formation for behavioral policy selection is intriguing – an example of how this works would be helpful (to ensure understanding of exactly how this might work – this seems relevant to earlier point about what is meant by “interoceptive adaptive behavior” as this could encompass many things at multiple levels. Much of the model lends itself to experimental predictions about how insular might interact with different parts of striatum in different situations.

Could these predictions be made more explicit, e.g. in a table ?

We apologize, following this revision the manuscript got a bit longer than expected, so, at this moment we avoided including this table. If the reviewer and editor consider this essential for the manuscript, then we will include a table in a future version.

P10, The section on dopamine and acetylcholine presumably relates to ACH in cortex (rather than striatum). in contrast to dopamine the relevant neuroanatomy (NBMeynert?) is not described earlier in the paper, nor illustrated in figure 2.

Figure 2 has been updated to reflect the density of acetylcholinergic receptors in each insula module by highlighting the thickness of the boxes describing the granular, dysgranular and agranular subregions and including a legend with this information.

It took two reads to understand how the proposal related to the generation and representation of feelings. The distinction between mesa and meta ception) does seem to cover this nicely – one assumes feelings are always present and changes draw attention to these at different functional levels Again the model is elegant in what is proposed; it’s parsing of different feeling states could be applied to understanding, for example alexithymia or disorders of empathy. The abstract suggests that this is relevant to conditions like addiction depression, PTSD and schizophrenia. This is not picked up later unfortunately. A compelling case was made recently regarding the metacognitive interoceptive basis to fatigue and depression. Although every disorder cannot be mentioned an IMAC account of fatigue and depression would be useful. With anxiety also linked to interoceptive prediction, how does the clinical expression of symptoms and behaviour fits within the model?

We deeply apologize to the reviewer for not describing possible roles of the IMAC in explaining psychiatric disorders in the first version of the manuscript. In the current version, we focused exclusively on how the IMAC architecture and functions could explain the development and prevention of depression. We mention fatigue as a somatic symptom that is frequently observed in depressive patients. In order to avoid redundancy with our previous works, we cited our work with Stephan on how fatigue and its relation with depression can be explained by allostasis and active inference (page 3, lines 25-29).

*To make our ideas clearer, we included a whole new section on how the IMAC model may explain the emergence of depression. See section 6 **IMAC Implications for understanding Depression.** The last paragraph under this section also highlights our unpublished findings demonstrating that abnormalities in the structure of the insula and other visceral neural systems (ACC, amygdala, hypothalamus) and that a possible disturbance of sympathetic control is associated with volume reduction of the bilateral posterior granular insula.*

REVIEWER: 2

Main suggestions

- The proposed insula lobular model of interoception (and specifically the “Body-brain interoceptive pathways” section beginning on page 3) can be better grounded in pre-existing theories of neurovisceral integration that outline the hierarchical, autonomic regulatory loops below the level of the insular cortex (See Smith et al 2017, “The hierarchical basis of neurovisceral integration”, *Neurosci Biobehav Rev*, DOI: 10.1016/j.neubiorev.2017.02.003 for an outline of the hierarchical functional anatomy). This can help establish the level at which the insula acts, as well as the nature of information it receives. The use of terms such as “information [passing]” (page 5, line 7) insinuates passive relaying of information, without adequate mention to the filtering, integration, or interpretation of information that occurs below the level of the insula, or that which is affected by descending modulatory input above the level of the insula. Finally, the predictive coding model states that prediction errors at these lower levels are what will propagate upwards, not just passive information relay of sensory information per se.

We are very grateful for this comment as in the submitted version of the manuscript we failed to explicitly identify the differences between lower level interoceptive systems located in the brainstem and higher level interoceptive systems, such as the insular cortex. We have added several sentences to highlight such differences, although we kept the main focus on the insular cortex. Our future work, now in preparation, will address in more details low hierarchical representations located in the brainstem as well as those located in the actual visceral organs, such as pacemaker cells in the heart and in the gut.

The second paragraph of the Introduction highlights, briefly, the hard-wired visceral and hormonal control exerted by multiple brainstem nuclei and also poses the question of why the cerebral cortex needs an insular interoceptive representation given that the brainstem is capable of generating nearly all visceral and physiological signals that control and maintain the body’s survival functions. Thus, the main goal of the manuscript is to suggest candidate roles of the insula in the processing of interoceptive information.

TEXT (page 2, lines 14-32 and following page): In the brain, the brainstem is the region that receives direct interoceptive information ascending from the visceral systems [1]. The brainstem contains several nuclei that receive direct ascending interoceptive information, such as nucleus tract solitary (NTS), the medullary reticular formation, the parabrachial nucleus (PBN), the periaqueductal gray area (PGA), and that are thought to generate innate, hard-wired, visceral and hormonal responses to maintain all physiological survival functions and to cope with the demands imposed by the body and the environment [1,12]. In the cerebral cortex, the insula and anterior cingulate cortex (ACC) are the main targets of interoceptive afferents arriving from the visceral systems through thalamic nuclei [3,5,13–17]. However, while the ACC is a predominantly agranular structure, i.e., it lacks a granular layer IV, the insula has a more complex organization with a topographic representation of visceral processes and three sub-regions with distinct levels of laminar granularity, distribution of acetylcholinergic receptors, and patterns of local, cortical, subcortical and brainstem connectivity [13,17–19] (Figure 1). These cytoarchitectonic and anatomical connectivity features suggest a central position and specialization of the insular cortex in processing interoceptive information. This convergence of cortical and interoceptive information upon the insular cortex has led to influential hypotheses concerning its potential roles in interoceptive prediction [14], information integration for awareness [20,21], emotional awareness [22], and interoceptive inference and emotion [23,24]. Despite the elegance and appeal of these hypotheses, no one has yet explained why the brain needs a cortical insular representation of visceral processes, given

that the multiple brainstem nuclei are sufficient to generate all necessary visceral and physiological adjustments to maintain the body's survival functions. Furthermore, it remains unclear how insular functions interact with—and are modulated by—input from cortical and subcortical systems and from neuromodulatory systems, and whether such interactions underwrite survival.

In following paragraphs still in the Introduction, we present the roles that may differentiate the interoceptive representations located in the insular cortex from those hard-wired representations located in the brainstem.

TEXT (page 3, lines 7-15): In this opinion piece, we turn to studies of insula cytoarchitectonic organization, neuroanatomical connectivity and recent theoretical formulations of brain function, such as allostasis, predictive coding, and active inference, to put forward the Insula Hierarchical Modular Adaptive Interoception Control (IMAC) model. The IMAC model proposes that the hierarchical and modular organization of the insular cortex—supported by its reciprocal connections with the prefrontal cortex (PFC) and the striatum, and modulated by the dopaminergic and acetylcholinergic systems—mediates (i) context-dependent control and learning of visceral responses, and (ii) the higher-order representation of conscious interoceptive feelings, which are built upon basic emotions and underlying visceral processes.

TEXT (page 3, lines 17-32; page 4, lines 1-6): According to the theory of active inference, the brain uses internal generative models, acquired through experience, to continuously generate descending or top-down predictions of expected sensory data [25–27]. In active inference, the goal of the agent is to find optimal policies that minimize free-energy, or prediction errors, between predicted and actual sensory input generated by the agent's interactions with, or sampling of, the environment [28,29]. The theory of allostasis proposes a similar predictive process for regulation of physiological states of the body [7,30–32]. The allostasis model suggests that the brain uses innate or learned prior knowledge of physiological states, e.g., glucose levels and heart rate, to predict future physiological states and thereby pre-emptively preclude deviations or prediction errors from homeostatic setpoints. Based on the allostatic reading of active inference [23,33,34], the IMAC model hypothesizes that the hierarchical modular cytoarchitecture of the insular cortex—supported by its parallel neural networks with the PFC and striatum-dopaminergic and acetylcholinergic systems—is specialized in higher-order interoceptive inference, hereby called metaception. That is, it specializes in the construction of cortical representations of lower-order, innate, interoceptive representations, herein called mesaception, located in subcortical, i.e., amygdala and hypothalamus, and brainstem nuclei. In the IMAC framework, metaception is regarded as an evolved cortical capacity to generate flexible higher order interoceptive predictions that are concurrently computed with action predictions that seek to maximize an individual's long-term fitness in her interactions with the environment. In other words, metaception is the perceptual synthesis (c.f., Bayesian belief updating) that furnishes high-level representations (c.f., feelings), which can predict lower level interoceptive representations and accompanying responses in the motor and autonomic domain.

TEXT (page 4, lines 8-14): The main premise of the IMAC model is that adaptive behavior—of animals and humans—revolves around acquisition of behavioral policies, not only for stimulus-action mapping, but also for concurrent generation and learning of associated visceral responses. To foreshadow the conclusions below, IMAC conceives of interoception as analogous to motor control, where action is realized by motor (resp. autonomic) reflexes that resolve proprioceptive (resp. interoceptive) prediction errors. These reflexes depend upon descending predictions or setpoints that are elaborated in deep hierarchical structures over increasing temporal scales.

We were unaware of the work by Smith but his paper was cited on page 20, lines 15-27.

- Organization of paper can be adjusted so that the functions of parallel networks linked with modules of the insula are introduced at the same time as merely describing the anatomy (“insula-cortical connections” and “insula-striatum-dopaminergic connections” sections). This would add flow to the paper, and help draw clearer links between the proposed functions and the neuroanatomy underpinning them. For example, the insula-cortical connections are only minimally mentioned early on in the paper, and the role of the PFC in interoceptive introspection is not adequately introduced.

We have reorganized the manuscript as much as possible to fulfill the reviewer’s request. We believe that the flow of the ideas presented in the manuscript have improved following the suggested changes. Following the suggestion by the other reviewer, we also extensively added references to support that modular organization of the insula based on its local cytoarchitecture, cortical, subcortical, thalamic and brainstem anatomical connectivity, topographic representation of visceral, cognitive and emotional information onto the insula structures.

- The following important terms are used often but not explicitly defined, and thus warrant specific attention. The authors could consider having a separate glossary for these terms.
 - o Interoceptive inference: see Seth & Friston (2016), “Active interoceptive inference and the emotional brain”, *Philos Trans R Soc Lond B Biol Soc*, DOI: 10.1098/rstb.2016.0007.

A sentence was added to suggest that the concept of active inference has been extended to the domain of interoception or active interoceptive inference.

TEXT (page 3, lines 27-31): Recent works have unified the concepts of active inference and allostasis under the umbrella of active interoceptive inference to suggest that the brain also creates generative models of the internal milieu of the body and uses such internal models to predict the desired states of the visceral organs and other physiological processes, such as heart rate, hormone release, activation of the immune system and energetic metabolism [10,14,23,33,34]

- o Interoceptive responses & control: Is this referring to the efferent output, mapped to certain levels/patterns of afferent input, that acts to regulate the present or future sensory interoceptive states?

*Overall, we were use the term **interoception** to reflect both sensation and control. In the revised version we removed this contradiction and added the term **visceral responses** to reflect changes in the functioning of the visceral organs that are generated as a consequence of interoceptive predictions.*

- o Higher-, intermediate-, vs. lower-order predictions: is the only difference between those the insular subregion from which they are generated? What relation does this distinction have with the mesa- (cortical) and meso- (subcortical) conceptions stated earlier?

The within insula hierarchy, we proposed in the manuscript, is an extension of the hierarchy observed in the PFC. We assigned the agranular insula the top of the hierarchy based on its connections with the VMPFC and APFC; the dysgranular insula was assigned an intermediate hierarchy based on its connections with the DLPFC; and the granular insula is the lower hierarchy based on its connections with the SMA and M1. The within insula hierarchy relates to metaceptions or cortical insular interoceptive representations. Mesaceptions are subcortical and brainstem interoceptive representations.

o See Figure 4 from Seth & Friston 2016, “Active interoceptive inference and the emotional brain” (Philos Trans R Soc Lond B Biol Sci, doi: 10.1098/rstb.2016.0007) who include other structures such as amygdala, ACC, and OFC but do not include lobular structure of the insula. These authors also do not have the distinction between sub-cortical “emotions” and cortical “feelings”.

The work by Seth and Friston is pioneer in proposing a general mechanism on how the cerebral cortex, in collaboration with the amygdala, hypothalamus, PBN and PAG, may implement active interoceptive inference.

Our work here can be considered an extension of the Seth and Friston model but with several additions. We concentrated on trying to explain why the cerebral cortex needs insula interoceptive representations, given that the brainstem can generate all basic signals to control visceral and physiological processes that main the body’s survival functions. In this regard, we suggest that not only the insula modular cytoarchitectonic organization but also their parallel connections with the prefrontal cortex may be involved in the formation of higher-order interoceptive representations to resolve interoceptive problems that can’t be realized in the brainstem.

Another addition is that brainstem interoceptive representations may give rise emotions related to basic survival drives, such as fear, hunger, thirsty, pain. These emotions seem to be represented in different brainstem nuclei with distinct ascending connections that may reach the insular cortex. However, these emotions may be felt without the need to reach the insula. On the other hand, when such emotions reach the insula, they may have at least three levels of hierarchy based on the three insula modules and that may give rise to conscious feeling and their meanings based on the insula connections with the prefrontal cortex (SMA, DLPFC, VMPFC).

o See Figure 1 of Barrett & Simmons 2015, “Interoceptive predictions in the brain” (Nat Rev Neurosci, doi: 10.1038/nrn3950), who do include hierarchical organization insular subregions but with a slightly different organization, and with cell bodies of precision cells residing in the insula itself.

Recent models of cortical function have suggested that some precision computations may be implemented in the cortical layers, including the work Barrett and Simmons, Bastos and Friston, and Doya. In the current manuscript, we avoided this topic and focused on how precision signals may computed on the insula-striatum-dopaminergic loops as an extension of the well-known precision (value) signals computed in the PFC-striatum-dopaminergic loops.

On the section Evidence Supporting the IMAC Model, we also highlight the idea of insula laminar precision computation proposed by Barrett and Simmons and suggest that future work may elucidate these differences.

TEXT (page 20, lines 4-24): At the neuromodulatory level, we suggested that the dopaminergic system computes the confidence (i.e., precision) of interoceptive predictions arriving from the insular cortex onto the striatum, in much the same way as it computes confidence about actions and decisions represented in the PFC-striatum network [136,142,172,175]. In contrast, it has been suggested that confidence signals are estimated at the laminar level in the insular cortex [14]. Future work needs to establish the exact differences in the confidence signals computed at the insula laminar structure and the insula-striatum-dopamine network. The dopaminergic system also has direct projections to the insula (Figure 3C) [169,254,255]. Precise roles of dopaminergic-insula connectivity have yet to be established, such as whether this pathway implements similar functions of synaptic plasticity and cognitive modulation hypothesized for the dopaminergic-PFC pathway [256,257].

o Additionally, the caption of this figure could be more clear in that (a) is not necessarily incorrect, but instead just more simplified than (b) which expands upon it.

Several improvements have been made to the figures.

- There is a large focus through the paper on the autonomic nervous system and visceral interoceptive processing. However, there are other pathways of interoception not mentioned in the paper that can play a role in many of the interoceptive and regulatory processes, emotions, and feeling states that are highlighted. Examples of these pathways include central chemoreceptors and hormones carried in the blood to the central nervous system (see Berntson & Khalsa 2021, “Neural circuits of interoception”, Trends in Neuroscience, DOI: 10.1016/j.tins.2020.09.011)

The work by Berntson and Khalsa is strong contribution to the literature. However, since it remains largely unknown how chemical interoceptive information is represented onto the insular cortex, we focused predominantly on the well-known insula interoceptive representations identified by neural fibers. We added, however, a few sentences to suggest the existence of chemical pathways.

TEXT (page 5, lines 5-9): Molecular interoceptive information, including nutrients (e.g. glucose), transport of blood gases (oxygen and carbon dioxide) and concentration and regulation of ions in neural tissue (e.g. potassium, sodium), which allows the maintenance of neural homeostasis (e.g. synaptic plasticity, development and preservation of neural structure), also reaches the brainstem and insular cortex through the vascular system and the blood-brain barrier [3,36–38].

TEXT (page 5, lines 22-25): The NTS receives significant chemical interoceptive signals from the area postrema (AP), a circumventricular organ which lacks a blood-brain barrier and senses chemical substances in the cerebrospinal fluid and the circulation (e.g., hormones, immune molecules) that modulate visceral functions and behavior [12,40].

- Instead of just listing the three terms, the title of the paper could be more cohesive and descriptive of the proposed relationship between them. For example, “feeling representation” is not mentioned until nearly the end of the paper, yet it is stated in the title. Additionally, although the title of the model (IMAC) emphasizes the modular architecture of the insula, it then de-emphasizes the other brain structures that play a role in interoceptive control and the visceral regulation.

Following the reviewer’s suggestion, we change the title of the manuscript to: An Insula Hierarchical Network Architecture for Active Interoceptive Inference.

- The authors could more clearly state which aspects of the model lack experimental evidence, while at the same time offering ways in which parts of this model could be directly tested using experimental tasks or manipulations. For example, how could researchers design studies to investigate the three specific levels of interoceptive processing in the subregions of the insula (interoceptive habit, model-based, vs. exploration, or mesa- vs. meso-conception)? Furthermore, how could dysfunction in these distinct processes be linked to what is known of different mental health disorders? This latter point will also help increase the clinical relevance of adopting such a model and its distinctions, and how this model could help better understand

disease processes, improve treatments, and predict patient outcomes.

A whole new section was added to suggest how the IMAC model may explain the emergence of depression (IMAC Implications for Understanding Depression). The ideas developed here are built upon our previous work on fatigue, as somatic symptom, and its link with depression.

Another new section (Evidence Supporting the IMAC model) was added to highlight some studies that support several aspects of the IMAC model as well as differences with previous models.

Here, we apologize to the reviewer but since the manuscript got longer than expected, we decided not to address what experimental designs could be used to test the ideas developed here.

Minor suggestions

- The concept of allostasis should be mentioned earlier on in the paper, when discussing “adaptive interoceptive inference” and the active inference framework (Figure 2)

A sentence on allostasis was added next to the one on active inference to reflect the similarity of these concepts.

TEXT (page 3, lines 18-26): According to the theory of active inference, the brain uses internal generative models, acquired through experience, to continuously generate descending or top-down predictions of expected sensory data [25–27]. In active inference, the goal of the agent is to find optimal policies that minimize free-energy, or prediction errors, between predicted and actual sensory input generated by the agent’s interactions with, or sampling of, the environment [28,29]. The theory of allostasis proposes a similar predictive process for regulation of physiological states of the body [7,30–32]. The allostasis model suggests that the brain uses innate or learned prior knowledge of physiological states, e.g., glucose levels and heart rate, to predict future physiological states and thereby pre-emptively preclude deviations or prediction errors from homeostatic setpoints

- Page 8 line 18: “neural” should be “neural structures”
This has been fixed.

- Page 11 line 20: “effect” should be “affect”
This has been fixed.

Optional suggestions

- The authors consider efferent, regulatory output under the definition of interoception, but could consider commenting on the distinction between afferent sensory and efferent regulatory arms.

The previous version of this manuscript tried to elaborate on the afferent sensory and efferent motor components of the autonomic nervous system. However, the details of these two branches are very complex and caused us a bit frustration due to the first author inability (who is writing this response) to clearly elaborate on their processes. Due to this inability, I (the first author) have decided to focus predominantly on the ascending interoceptive pathways to the insular cortex. Figure 1B was added to show multiple parallel ascending interoceptive pathways from the head and the body to the insula. Figure 1A, however, was kept to show to the audience the motor effects of the sympathetic and parasympathetic pathways on visceral control.

- The introductory paragraph to begin the paper reads more like a list, and never defines interoception.

We have rephrased the second sentence of the first paragraph to more explicitly provide a definition of interoception.

TEXT (page 2, lines 4-8): Recent studies have sought to understand how such highly complex psychological processes—and pathological states, e.g., depression—are influenced by interoception, the sensation of information ascending to the brain from visceral systems and physiological processes under the control of the autonomic nervous system (ANS) [2–11].

- Other cortical areas in addition to the insula can be outlined more fully, in relation to the processes they contribute to, rather than kept in one separate section towards the end. For example, other areas such as the cingulate cortex may contribute to the generation of visceromotor commands that can occur as part of motor, affective, and cognitive networks for the regulation of internal bodily states.

We appreciate this suggestion and believe the reviewer's comments on the ACC is well put. However, in the present manuscript, we are basically advocating for a similar function to the insular cortex. Furthermore, the ACC has been associated with multiple cognitive functions, such as value computation, confidence estimation, resolution of uncertainty, cognitive and behavior switching. How to integrated these different views on the ACC in order to explain its precise roles in visceromotor control has yet to be established. Also, what differentiates the visceromotor commands of the insula and ACC remain unclear. We are currently engaged in understanding the differences between these two neural systems and this will appear in our future work.

For other brain regions whose functions are well-established, we attempted to briefly suggest their candidate roles to the insula interoceptive functions. This view appears in the last paragraph of the section Insula Hierarchical Modular Adaptive Interoception Control.

TEXT: Other neural systems also contribute to acquisition of interoceptive representations, given their prominent direct or indirect connections with the insular cortex. For instance, the TPJ, the amygdala, hippocampus, and anterior cerebellum are well known for their participation in social cognition, classical conditioning, episodic memory and encapsulation of sensory-motor mappings, respectively [127,156–158]. The IMAC model hypothesizes that the TPJ may contribute to the formation of social interoceptive predictions (e.g. visceral or physiological response in social relations) [159], the hippocampus may contribute to acquisition of episodic interoceptive predictions [160], the amygdala may contribute to formation of Pavlovian interoceptive predictions [161], and the anterior cerebellum, which has neuroanatomical loop connections with the SMA [162,163] and is implicated in sequential learning and acquisition of forward models [119,127,132], may also be able to generate encapsulated interoceptive predictions or forward interoceptive predictions to produce automatic sequential visceral responses [119,127,158].

- In Figure 3, the authors could consider adding the dopaminergic areas that modulate the precision of signals.

Figure 3 as updated with the addition of a panel C which highlights a few parallel ascending interoceptive pathways, including the amygdala, habenula, dopaminergic and serotonergic systems,

previously implicated in depression.

Appendix B

Associate Editor Comments to Author (Dr Anastasia Christakou):

Associate Editor

Comments to the Author:

Alongside the well detailed suggestions of the reviewer, I strongly encourage you to pay particular attention to interventions that will make the paper accessible to a wider readership, such as clarifying (your use of) specialist or ambiguous terms and editing the schematic depiction of the circuitry in figure 1B (e.g.: you may wish to sacrifice detail for clarity of communication on the basis of the message that the figure is designed to support). I would also encourage you to favour clarity and completeness over brevity when considering your treatment of the comments about clarifying the model's relationship with existing models, and providing examples of testable predictions/falsifications.

Response

We are very grateful to the editor and reviewers for their suggestions and comments. We have tried to respond all your concerns and implement your suggestions as much as possible.

We strongly believe that changes in the text, clarification of terms, addition of examples, and clarity of the figures contributed tremendously to improve the quality of the manuscript.

Reviewer comments to Author:

Reviewer: 2

COMMENT

- Although the "Evidence for the Model" section is a great addition to the paper, the authors could consider distributing it throughout the paper under the corresponding subsections rather than leaving it all to the end. For example, the anatomical modular structure of the model could then be followed by evidence for this anatomical arrangement, and so on.

Response

We are very grateful for the suggestion to present the evidence for the proposed throughout the manuscript. We have done that and realized that it improves immensely the quality of our work.

COMMENT

- Although implied, the authors could explicitly state the intuitive (and theoretically necessary) link between actions – i.e. the somatomotor movements that constitute behaviours – and the visceromotor responses they require for energetic support. Somatomotor commands to skeletal muscles must be coupled with the appropriate visceromotor changes in body parameters to support blood flow, blood glucose, etc. Although these changes could happen retrospectively via reflexive autonomic loops, it is likely that the brain circuits responsible for the initiation of actions must also initiate the prospective interoceptive predictions that generate these visceral responses. This pairing of somatomotor and visceromotor outputs could be emphasized.

Response

We understand that we failed to more clearly emphasize how actions and interoceptive responses interact. We have added the following sentences with examples on action-interoception mapping.

Page 5, Line 4-15.

The main premise of the IMAC model is that adaptive behavior of animals and humans revolves around acquisition not only of action policies, such as stimulus-action mappings or social interaction strategies that maximize rewards, survival and reproduction [39,40], but also for acquisition of interoceptive policies that are needed to either maintain the body's physiological and visceral survival functions and concurrently to support mental processes and implementation of action policies, eventually leading to mapping or binding of action-interoception policies. Take the wake-sleep cycle as a simple example of an action-interoception policy mapping in which the brain uses interoceptive policies to generate visceral responses appropriate for wake-sleep behaviors: humans display higher blood pressure, increased heart rate, fast metabolism, reduced melatonin production and increased cortisol release in the waking period when the level of physical activity is higher than in the sleep period, when the physiological demands of the body are highly diminished [41].

Page 12, Line 17-32; Page 13, Line 1-2

As an example of this action-interoception mapping, we have cited earlier the physiological changes observed in humans during wake-sleep cycles, such as changes in activity of neuromodulatory systems in the brain, e.g. higher serotonin and lower acetylcholine, increased heart rate, increased energetic metabolism, increased body temperature, increased respiration, and decreased plasma melatonin during the wake period than during the sleep period [41,167,168]. During exercise, relative to the resting state, there are numerous physiological responses generated by visceral systems to support muscle performance, such as increased consumption of oxygen, increased cardiovascular, hormonal, metabolic, sweating and thermal regulatory responses [169–172]. There also are many daily situations in which the body generates physiological and visceral responses, such as increased heart rate, blood pressure, skin conductance, pupil dilation, in anticipation to aversive or reward predicting cues, public speech, social interactions and physical exercise [134,173–175]. Adaptive cardiovascular responses in humans are also observed in space flights, subaquatic diving, profession type and season of the year, in athletes of different sport modalities, and in response to the demands of cognitive and emotional tasks [176–180]. Without such visceral and physiological adjustments, it would be impossible to perform successful movements, to have a good night of sleep, to react appropriately to demands of the environment or to prepare the body and plan behaviors to anticipated stress.

COMMENT

- While the central question of the paper is strong (why is a cortical representation of bodily signals necessary when the subcortical and brainstem structures can support survival?), the authors could better comment on whether conscious feelings are necessary for the higher-order interoceptive representations and policies. Theoretically, metaception and the associated prospective predictions could occur without subjective feelings of the body states, and much of this body regulation happens normally below the level of conscious awareness as it is (e.g. most people are not aware of their heartbeat under resting conditions). However, it could be that conscious feelings are a useful tool to promote the regulation of body states in addition to visceral responses (like putting on a sweater when cold before instigating shivering responses).

RESPONSE

We are grateful for this comment and we understand that in the previous version of the manuscript the link between feelings and their roles in behavior were unclear. We have inserted a few sentence to clarify this idea.

Page 20, Line 18-23

For instance, in a situation in which someone experiences an aversive event that leads to an increase in heart rate, the insula will send interoceptive prediction errors to the PFC, which may interpret, categorize and contextualize them as a speeding heart associated with an imminent threat or a simple change of body posture, or after having a meal, the PFC may signal whether uncomfortable stomach signals indicate an unpleasant meal or an overly distended stomach caused by overeating.

Accordingly, the lower-order gINS-SMA network may generate implicit or habitual metaceptions commonly associated with “gut” or intuitive feelings that may be subpersonal and may be acquired after repeated experiences and may support the implementation of habitual action-interoception policies in response, for instance, to emotionally salient events, e.g., quickly escape from a snake attack or protect oneself from a sport injury. In contrast, the higher-order dINS-DLPFC and aINS-VMPFC networks may contribute to representations of explicit, introspective, conscious feelings that contribute to understand one’s physiological and visceral states in the early stages of learning of novel experiences or to reason and implement action-interoception policies, e.g., behavioral decisions, visceral and physiological responses, that solve current or future demands of the body and environment, e.g., cooking a meal in anticipation of high hunger or turning on the air conditioner to decrease indoor and body temperature. These higher-order conscious feelings may also be used generate action-interoception policies that solve lower-order interoceptive representations and associated interoceptive prediction errors arriving from the gINS-SMA network or from the brainstem;

COMMENT

- The authors could more directly address how this model compares with other theories and its importance within the literature, specifically the EPIC model proposed by Barret and Simmons (Nature Reviews Neuroscience 2015). This could be done by stating how the IMAC model is similar in terms of the emphasis on cytoarchitecture differences, but expands upon the EPIC model by including broader networks of brain regions. It could also draw upon some of the cytoarchitectonic detail of the EPIC model in terms of the cell types and information passing within the three main subregions.

RESPONSE

We have added and edited the following sentences that indicate some differences between our model and others.

Page 16, line 8-32; Page 17, Line 1-4

The insula active interoceptive inference neural architecture and hypotheses put forward above address several missing issues left unexplained in previous models of insula function. For instance, previous models using predictive and error-correction approaches [14,25] or information integration [20,21] sought to suggest specialized functions for the insula, based on how its local architecture processes a multitude of visceral, cognitive and emotional inputs it receives and its activation across multiple task domains. In contrast, the IMAC model assigns interoceptive inference functions to the insula based supported by the parallel network connections it forms with the PFC and striatum and their well-known roles in adaptive behavior as well as the neuromodulatory input from dopaminergic and acetylcholinergic systems onto these networks. Thus, the IMAC framework can explain why neural activity of the insula is found in various emotional, motivational, social and cognitive tasks. Another difference in relation to previous models is our three-layer hierarchical architecture with first-order, second-order, and third-order interoceptive representations located in brainstem-subcortical systems, insular cortex, and PFC, respectively. Here, each hierarchical level is defined based on their intrinsic functional properties to generate innate autonomic reflexes or first-order interoceptive predictions, e.g., the brainstem and subcortical regions, or more flexible higher-order interoceptive predictions in the insula, e.g., second-order, and PFC, e.g., third-order. However, the organization of neural pathways connecting the ANS with the brain is more complex than a simple functional three-layered model (Figure 2) and has recently been recognized in the neuroanatomical eight-layer hierarchical neurovisceral integration (NVI) model [219]. Despite different numbers of hierarchical layers, the IMAC and NVI models apply the same principles of predictive coding and Bayesian belief updating to suggest how interoceptive representations emerge at each hierarchical level. Future models or updated versions of both IMAC and NIV models should consider in more detail how to define levels of hierarchical interoceptive organization based on the number of synaptic connections linking the visceral systems to the insular cortex, their innate or flexible interoceptive representations, the interaction between the parallel interoceptive pathways (Figure 3C), as well as the local cellular and molecular circuitry of the visceral systems, e.g., cardiac pace maker cells or the direct influences of hormones and other circulating chemical on visceral functions.

Page 18, Line 9-17

At the cellular level, the IMAC model provided above a mechanistic, but simplistic interpretation of how striatum direct and indirect pathway medium spiny neurons and their dopaminergic input may compute interoceptive confidence signals, i.e., precision, of insula descending interoceptive predictions (Figure 2), in much the same way as it computes confidence about actions and decisions represented in the PFC-striatum network [29,30,152,155,165,166,236,240,241]. In contrast, another model has suggested a mechanism for estimation of interoceptive confidence implemented by precision units located within the laminar structure of the insula [14]. Future work needs to establish the exact differences in the confidence signals computed at the insula laminar structure and the insula-striatum-dopamine network.

COMMENT

- The authors seem to conflate the terms emotion and feeling; for example, page 1 line 15 they mention the “higher-order representation of conscious interoceptive feelings, which are built upon the basic emotions and underlying visceral processes”. They never offer an explicit definition of an emotion versus the feeling of an emotion or overarching affect, and this is important given the highly debated definition of an emotion. Additionally, while they maintain that emotions are non-conscious, they state on page 15 line 1 that “feelings are under conscious and higher-order cognitive control”. While conscious control could affect a meta-awareness of the feeling, can it affect the generation of the feeling itself?

RESPONSE

We understand the reviewer’s concern on the absence of a clear definition of the terms emotion and feeling and the intense debate on the literature on this topic. Our goal was not to create more problems by inventing new definitions of emotion and feelings, so, we intentionally avoided this battle. As a compromise to the reviewer’s suggestion, we edit the following sentence and pointed the audience a few references on this topic.

Page 18, Line 28-30

In order to simplify our treatment, we will use here a general notion found in the literature that emotions are unconscious arousal states linked with visceral and physiological processes under reflexive control, and feelings are conscious representations of emotions [14,21,23,249–257].

Page 19, Line 5-8

These findings led to influential neuropsychological theories proposing that emotions arise from a combination of interoceptive signals triggered by physiological changes in the functioning of visceral systems and associated behavioral repertoires [210,249,250,256].

Page 19, Line 11-19

For example, low glucose and insulin afferent signals onto brainstem systems generate interoceptive prediction errors that activate a mesoceptive representation of the emotion of hunger and triggers homeostatic responses and food-specific consumption behaviors through a specialized neural pathway in the hypothalamus [259,262]. This interpretation that emotions arise from brainstem interoceptive prediction errors is consistent with previous proposals that emotions emerge from dynamics in the rate of change, increase or decrease, of free-energy or

interoceptive prediction errors triggered by visceral and physiological deviations from their expected functional parameters or setpoints [29].

Page 19, Line 26-27

There have been multiple suggestions that cortical brain regions, including the insular cortex, contribute to generation of conscious feelings emerging from emotions [21,22,249,265–270].

Page 20, Line 4-7

Then how does consciousness of feelings emerge from insula metaception? The IMAC model offers a specific hypothesis that consciousness of feelings and bodily states emerges from insula metaceptive representations and insula-PFC interactions, and is built up from experiences, innate emotional states, visceral and physiological responses associated with them.

Page 20, Line 14-23

The pattern of insula-PFC anatomical connectivity allows the PFC to form higher- or third-order interoceptive representations, and to use complex cognitive functions, such as introspection, to inspect, interpret, categorize, and reason on the contents, causes, and consequences of second-order interoceptive representations furnished by the insular cortex. For instance, in a situation in which someone experiences an aversive event that leads to an increase in heart rate, the insula will send interoceptive prediction errors to the PFC, which may interpret, categorize and contextualize them as a speeding heart associated with an imminent threat or a simple change of body posture, or after having a meal, the PFC may signal whether uncomfortable stomach signals indicate an unpleasant meal or an overly distended stomach caused by overeating.

Page 20, Line 30-32; Page 21, Line 1-15

Empirical findings, showing PFC-striatum networks engaged in distinct stages of learning, e.g., the DLPFC-mStr and VMPFC-vStr in early flexible-conscious learning and the SMA-pStr in late habitual-unconscious learning, suggest that even among insula-PFC parallel networks there may be hierarchical representations of conscious feeling. Accordingly, the lower-order gINS-SMA network may generate implicit or habitual metaceptions commonly associated with “gut” or intuitive feelings that may be subpersonal and may be acquired after repeated experiences and may support the implementation of habitual action-interoception policies in response, for instance, to emotionally salient events, e.g., quickly escape from a snake attack or protect oneself from a sport injury. In contrast, the higher-order dINS-DLPFC and aINS-VMPFC networks may contribute to representations of explicit, introspective, conscious feelings that contribute to understand one’s physiological and visceral states in the early stages of learning of novel experiences or to reason and implement action-interoception policies, e.g., behavioral decisions, visceral and physiological responses, that solve current or future demands of the body and environment, e.g., cooking a meal in anticipation of high hunger or turning on the air conditioner to decrease indoor and body temperature. These higher-order conscious feelings may also be used generate action-interoception policies that solve lower-order interoceptive representations and associated interoceptive prediction errors arriving from the gINS-SMA network or from the brainstem;

COMMENT

- The authors could provide a better definition of the following machine learning terms in the context of the IMAC model and how these terms apply to interoception: “model-based”, “interoceptive policies”, and “environmental state transitions”. What is the “goal of the agent” for interoceptive inference models? These are important aspects of the IMAC model and are mentioned numerous times, but if not familiar with machine learning literature I worry it may not be clear for some readers.

o Additionally, the authors mention “interactions with, or sampling of the environment” in their definition of active inference on page 3, but they do not mention what “sampling of the environment” would look like for the inner body environment, as in interoceptive inference? In active inference, agents can choose behaviours that seek out more information about the environment, e.g. picking up an object to get a better visual... for the inner body, would this be akin to gating mechanisms that amplify or suppress interoceptive input to insular cortices?

Response

In order to clarify some of these terminologies as pointed out by the reviewer, we have added several sentences, example and references to help the reader understand these terms.

Page 3, Line 24-31 (a quick definition of action policy and environment sampling)

In active inference, the goal of the agent is to find optimal action policies, e.g. rules or strategies for quick selection of actions, muscle activation patterns, decisions, and social behaviors in a given context, that minimize free-energy, or prediction errors, between predicted and actual sensory input generated by the agent’s interactions with, or sampling of the environment, e.g., quality of social interactions at home or in public, street navigation while driving or walking, selection of healthy food, learning to play a musical instrument, whether to dribble or pass the ball while playing basketball, an infant learning to walk on a slippery or rough surface [30,31].

Page 4, Line 4-20 (a elaboration on the goal of the agent in interoceptive inference, short definition of interoceptive policy, and examples of interoceptive sampling)

Recent studies have unified the concepts of active inference and allostasis under the umbrella of active interoceptive inference to suggest that the brain also creates and stores generative interoceptive models of the internal milieu of the body and uses such interoceptive models to explain ascending interoceptive signals and to generate descending interoceptive predictions to regulate and achieve desired states of the visceral organs and physiological processes, such as heart rate, hormone release, activation of the immune system and energetic metabolism [10,14,23,29,35,36]. According to active interoceptive inference approach, the goal of the agent and the brain is to find optimal interoceptive policies, e.g., visceral and physiological response patterns that can be quickly selected for implementation in a given context, that minimize interoceptive prediction errors between predicted and actual interoceptive input arriving from visceral and physiological systems. Interoceptive policies are acquired by sampling visceral responses occurring at a given time and context, e.g. heart rate and hyperventilation, e.g., lung inflation while running, breathing speed, stomach motility and pain after a meal, bladder dilation with urine production, and other physiological processes, e.g., decrease in glucose level with increased hunger, hormones released after physical activity or psychological stress, immune molecules and inflammatory processes following tissue stress, body temperature changes in a cold or hot day.

COMMENT

- The authors comment numerous times on topographic representation of interoceptive information within the insula (e.g. page 7, “posterior to anterior” “topographic viscerosensory maps”), while also maintaining the integrated nature of such information (“information integration for awareness”, page 2; “integration and relay of interoceptive information” in the brainstem, page 5; hierarchical neurovisceral integration model, page 20). However, they do not comment on the potentially contrasting nature of these ideas, and what this could mean for the nature of the information reaching the insula... topographic representation implies the preservation of more or less parallel streams of sensory information ascending through the brainstem/thalamus relays to the insula, whereas integration implies the combining of information at lower levels or within the insula modular hierarchy. There are some opinions against simple spatial topographic mapping, instead opting for pattern-based representations; see Avery Current Opinion in Physiology 2001 for an example using taste.

Response

We are grateful to the reviewer for raising the problem of topographic vs. distributed representation of physiological signals on the insular cortex. In order to simplify our manuscript, we elaborated our hypothesis based on the numerous neuroanatomical findings that point to the existence of parallel visceral pathways ascending from the brainstem and synapsing onto the insula and this was explicitly shown in Figure 1. We recognize that these neuroanatomical pathways are not entirely segregated and that interoceptive representation coding in the insular cortex may be more broad than specifically localized. We incorporated a brief sentence to express the reviewer's concern.

Page 9, Line 4-11.

Although here we use the multiple ascending interoceptive neuroanatomical pathways and their topographic representation onto the insular cortex as an important feature of the IMAC model, these pathways are not entirely anatomically segregated and non-overlapping in their insular representation. For example, human neuroimaging experiments have shown that insular representation of heart, stomach and bladder overlaps with representation of gustatory information [104–106]. The functional significance of overlapping neuroanatomical and functional insular interoceptive representations needs to be investigated in future studies.

COMMENT

- The figures have been improved for comprehensiveness and completeness, and Figures 1A and 1B do give the reader an indication of just how complex the interoceptive system is with all the different body systems, pathways, and brain regions/connections that are involved. However, Figure 1B still has too much detail that makes it visually confusing. With so many crossing arrows, it is hard to follow the connections and it's not clear which are wanting to be highlighted. The authors could consider better ways to combine, consolidate, and simplify some of the arrows, and if necessary split up the figure further into one overall figure with then sub-figures providing more detail on the connectivity of different parts (but not all the detail of all the parts on just one figure).

Response

We humbly admit that Figure 1B was extremely busy and almost impossible to understand. We tried as much as possible to implement the reviewer's suggestions to improve this figure. We decided to stick with a single figure, instead of creating subfigures. We, however, reorganized the figure by removing a large number of boxes and connecting arrows. To improve visualization of the connections some boxes were associated with specific colors and their respective arrows are also of the same color. Like in graph models of brain connectivity, in the new figure connections leave the node from the same point (indicated by a closed circle on the top of a node); arrows from different nodes also arrive on the same point of their target node. We believe that this reorganization brought a very strong improvement to the figure.

Comment

- The first couple sentences of the "IMAC Implications for Understanding Depression" seem slightly off-topic.. it may be better to begin by stating how interoceptive dysregulation is implicated in the pathophysiology of depression, before mentioning the public health importance of depression and its prominence given the COVID-19 pandemic.

Response

Thank you very much for the comment. We completely removed that awkward sentence from the text.

Comment

- The authors could consider citing Hassanpour et al. *Neuropsychopharmacology* 2018 or Teed et al. *Jama Psychiatry* 2022 when referencing tasks that deliver unpredictable interoceptive stimulation (page 21 line 23). These papers increased cardiorespiratory arousal during scanning via intravenous infusions of the adrenaline analogue isoproterenol, and found regions of the insula to be primarily responsive to this peripheral stimulation.

Response

We were unaware of those references that are now cited in the manuscript.